# Revisiting Nonstationary Kernel Design for Multi-Output Gaussian Processes

**Qiaochu Xu**[*,1] **Zi Yang**[*,1] **Ying Li**[†,1,2] **Michael Minyi Zhang**[†,1] **Pablo M. Olmos**[3]

[1] School of Computing and Data Science, The University of Hong Kong
[2] Department of Statistics and Data Science, National University of Singapore
[3] Department of Signal Theory and Communications, University Carlos III de Madrid
{qiaochu.xu, ziyang2023, lynnli98}@connect.hku.hk
mzhang18@hku.hk  ying-li@nus.edu.sg  pamartin@ing.uc3m.es

## Abstract

Multi-output Gaussian processes (MOGPs) provide a Bayesian framework for modeling non-linear functions with multiple outputs, in which nonstationary kernels are essential for capturing input-dependent variations in observations. However, from a spectral (dual) perspective, existing nonstationary kernels inherit the inflexibility and over-parameterization of their spectral densities due to the restrictive spectral–kernel duality. To overcome this, we establish a generalized spectral–kernel duality that enables fully flexible matrix-valued spectral densities — albeit at the cost of quadratic parameter growth in the number of outputs. To achieve linear scaling while retaining sufficient expressiveness, we propose the multi-output low-rank nonstationary (MO-LRN) kernel: by modeling the spectral density through a low-rank matrix whose rows are independently parameterized by bivariate Gaussian mixtures. Experiments on synthetic and real-world datasets demonstrate that MO-LRN consistently outperforms existing MOGP kernels in regression, missing-data interpolation, and imputation tasks. Code is publicly available at https://github.com/KrnteXu/MO-LRN.

## 1 Introduction

Gaussian processes (GPs) (Williams and Rasmussen, 2006) offer a principled Bayesian non-parametric framework that is both flexible and interpretable for modeling complex nonlinear functions. Extending GPs to the multi-output setting leads to a multi-output Gaussian process (MOGPs) (Alvarez et al., 2012), which, rather than modeling each output independently as in standard GPs, explicitly capture statistical dependencies across multiple outputs. This capability is particularly valuable in diverse application domains, such as early sepsis detection (Futoma et al., 2017), traffic speed prediction (Rodrigues et al., 2018), financial risk modeling (Nguyen et al., 2014), and heterogeneous-output modeling (Moreno-Muñoz et al., 2018).

Formally, an MOGP defines a priori over a vector-valued function $\mathbf{f} : \mathbb{R}^D \to \mathbb{R}^V$: $\mathbf{f}(\mathbf{x}) = \left[f_1(\mathbf{x}), \ldots, f_V(\mathbf{x})\right]^\top$, with zero mean and a matrix-valued covariance (kernel) function $\mathbf{K}(\mathbf{x}_1, \mathbf{x}_2) = \left[k_{ij}(\mathbf{x}_1, \mathbf{x}_2)\right]_{i,j=1}^V \in \mathbb{R}^{V \times V}$ (Alvarez et al., 2012), i.e.,

$$\mathbf{f}(\mathbf{x}) \sim \mathcal{GP}\big(\mathbf{0}, \mathbf{K}(\mathbf{x}_1, \mathbf{x}_2)\big).$$

Given a set of inputs $\mathbf{X} = [\mathbf{x}_1, \ldots, \mathbf{x}_N]^\top \in \mathbb{R}^{N \times D}$ (distinct from the generic $\mathbf{x}_1, \mathbf{x}_2$ above), the corresponding outputs

$$\boldsymbol{\mathcal{F}} = \big[\mathbf{f}(\mathbf{x}_1), \ldots, \mathbf{f}(\mathbf{x}_N)\big]^\top \in \mathbb{R}^{N \times V}$$

follow a joint Gaussian distribution:

$$p(\boldsymbol{\mathcal{F}}) = \mathcal{N}\big(\text{vec}(\boldsymbol{\mathcal{F}}) \mid \mathbf{0}, \boldsymbol{\mathcal{K}}(\mathbf{X}, \mathbf{X})\big),$$

---

[*] Equal contribution. [†] Corresponding author.

where

$$\mathcal{K}(\mathbf{X}, \mathbf{X}) = \left[ \mathbf{k}_{ij}(\mathbf{X}, \mathbf{X}) \right]_{i,j=1}^{V} \in \mathbb{R}^{NV \times NV}$$

is a block-structured covariance matrix where each element of the block $[\mathbf{k}_{ij}(\mathbf{X}, \mathbf{X})]_{pq} = k_{ij}(\mathbf{x}_p, \mathbf{x}_q)$. Intuitively, the diagonal blocks $\mathbf{k}_{ii}(\mathbf{X}, \mathbf{X}) \in \mathbb{R}^{N \times N}$ capture output-specific patterns, while off-diagonal blocks $\mathbf{k}_{ij}(\mathbf{X}, \mathbf{X}) \in \mathbb{R}^{N \times N}$, $i \neq j$, capture cross-output dependencies. Note that when $V = 1$, the above definition reduces to the standard GP (named single-output GP (SOGP) for convenience) with scalar kernel $k(\mathbf{x}_1, \mathbf{x}_2)$. Using a standard GP for $V$-dimensional outputs assumes independence across dimensions, i.e., $p(\mathcal{F}) = \prod_{i=1}^{V} \mathcal{N}(\mathcal{F}_{:,i} \mid 0, \mathbf{k}_{ii}(\mathbf{X}, \mathbf{X}))$, which is equivalent to $\mathcal{K}(\mathbf{X}, \mathbf{X})$ being block-diagonal.

A key aspect of GPs is the choice of the covariance function, as it encodes prior assumptions about the latent functions $\mathbf{f}$, which in turn determine the patterns the model can capture in the data. Stationary kernels are the most prominent choice (Ulrich et al., 2015; Parra and Tobar, 2017; Alvarez and Lawrence, 2008), where covariance depend only on relative distances $\mathbf{x}_1 - \mathbf{x}_2$. Consequently, an MOGP model with such a kernel cannot capture input-dependent variations in the observations which are prevalent in real-world data.

Thus, designing nonstationary kernels for MOGPs is essential. A straightforward approach is to extend the linear model of coregionalization (LMC) by replacing its SOGP base kernel with a nonstationary variant (Boyle and Frean, 2004; Álvarez et al., 2010). However, from a spectral (dual) perspective, this extension still restricts the form of cross-output covariances, thus limiting flexibility (Parra and Tobar, 2017). Recently, Altamirano and Tobar (2022) introduced the multi-output harmonizable spectral mixture (MOHSM) kernel, which constructs a two-level mixture spectral density and maps it to the kernel domain via an existing duality to overcome this problem. Nevertheless, because the duality enforces structural conditions on the spectral density, its design ends up over-parameterized but still inflexible. These limitations carry over to the kernel, reducing its ability to model nonstationarity and creating a redundant parameter space that is difficult to optimize.

In this paper, we propose an expressive and parameter-efficient nonstationary kernel for MOGPs. Our contributions are as follows:

1. We establish a new duality between the spectral density and the kernel, removing conventional restrictions and thereby enabling, in theory, fully flexible matrix-valued spectral densities, though with quadratic parameter growth in $V$.

2. We propose both a parameter-efficient and sufficiently expressive multi-output low-rank nonstationary (MO-LRN) kernel by first specifying a low-rank spectral density with independent bivariate Gaussian-mixture factors to reduce parameter growth to linear in the output dimension, and then mapping it back to the kernel space via the new duality.

3. We provide experimental results covering tasks such as regression, missing-data interpolation, and imputation, on both synthetic datasets and diverse real-world benchmarks, to show that MO-LRN achieves state-of-the-art (SOTA) performance compared with all existing MOGP kernels.

## 2 BACKGROUND AND PROBLEM STATEMENT

In this section we examine the design of the MOHSM kernel by taking the next-gen spectral mixture (NG-SM) kernel, which is currently the leading nonstationary kernel for SOGPs, as a reference point. We first present NG-SM and MOHSM (§ 2.1, § 2.2), and then revisit the MOHSM design through the lens of the NG-SM's principles, highlighting its key shortcomings via theoretical analysis and empirical validation.

### 2.1 NEXT-GEN SPECTRAL MIXTURE (NG-SM) KERNEL

The NG-SM kernel is a single output nonstationary kernel that is theoretically capable of approximating any continuous kernel arbitrarily well (Yang et al., 2025). Its core design principle is to specify a dense spectral density and subsequently derive the corresponding kernel function via the well-established duality between kernels and spectral densities. Formally, this duality is characterized by the following theorem:

**Theorem 1** (Universal Bochner's Theorem). *A complex-valued bounded continuous kernel $k(\mathbf{x}_1, \mathbf{x}_2)$ on $\mathbb{R}^D$ is the covariance function of a mean square continuous complex-valued random process on*

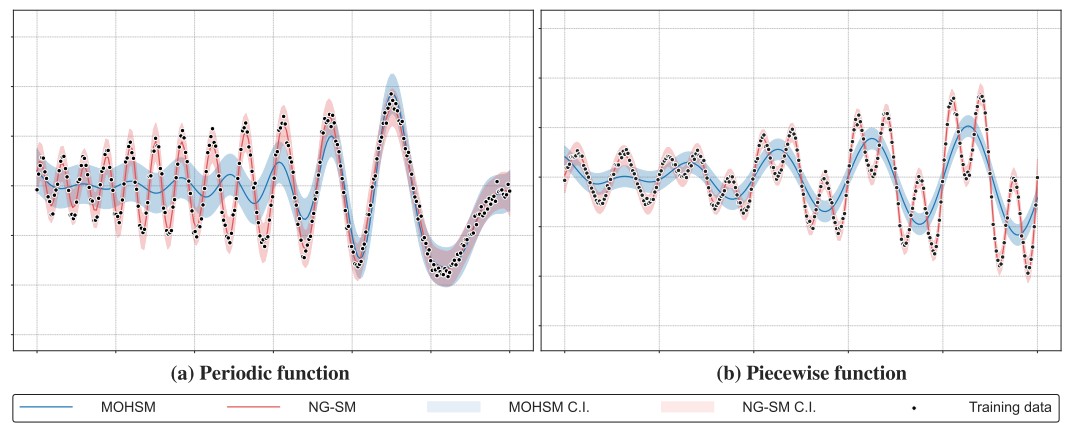

**(a) Periodic function**     **(b) Piecewise function**

| —— MOHSM | —— NG-SM | MOHSM C.I. | NG-SM C.I. | • Training data |

**Figure 1: MOHSM vs. NG-SM fits on scalar observations.** Panel (a) shows a periodic function, and panel (b) a piecewise function. In both panels, the blue line denotes the MOHSM posterior mean and the red line the NG-SM posterior mean. The shaded light blue and pink regions indicate the corresponding 95% confidence intervals. Black dots mark the training data.

$\mathbb{R}^D$ *if and only if*

$$k(\mathbf{x}_1, \mathbf{x}_2) = \frac{1}{4} \int \Big[ \exp\big(i\boldsymbol{\omega}_1^\top \mathbf{x}_1 - i\boldsymbol{\omega}_2^\top \mathbf{x}_2\big) + \exp\big(i\boldsymbol{\omega}_2^\top \mathbf{x}_1 - i\boldsymbol{\omega}_1^\top \mathbf{x}_2\big)$$
$$+ \exp\big(i\boldsymbol{\omega}_1^\top (\mathbf{x}_1 - \mathbf{x}_2)\big) + \exp\big(i\boldsymbol{\omega}_2^\top (\mathbf{x}_1 - \mathbf{x}_2)\big) \Big] u(d\boldsymbol{\omega}_1, d\boldsymbol{\omega}_2), \tag{1}$$

*where $u$ is the Lebesgue-Stieltjes measure associated with some function $p(\boldsymbol{\omega}_1, \boldsymbol{\omega}_2)$. When $u(\omega_1, \omega_2) = \delta(\omega_1 - \omega_2)\, u(\omega)$ (with $\omega \triangleq \omega_1 = \omega_2$), this theorem reduces to Bochner's theorem.*

By specifying the spectral density as a mixture of bivariate Gaussians, i.e., $p(\boldsymbol{\omega}_1, \boldsymbol{\omega}_2) = \sum_{q=1}^Q \alpha_q s_q(\boldsymbol{\omega}_1, \boldsymbol{\omega}_2)$, where each $s_q(\boldsymbol{\omega}_1, \boldsymbol{\omega}_2)$ is a bivariate Gaussian component, the NG-SM kernel can be derived via Eq. (1). Leveraging the fact that Gaussian mixtures are dense in the space of continuous functions (Plataniotis and Hatzinakos, 2000), the NG-SM kernel can approximate any continuous kernel arbitrarily well.

## 2.2 MULTI-OUTPUT HARMONIZABLE SPECTRAL MIXTURE (MOHSM) KERNEL

The current exemplar for nonstationary multi-output kernels is the MOHSM kernel (Altamirano and Tobar, 2022). It follows the general paradigm of specifying a matrix-valued spectral density and mapping it back to the kernel domain via a multivariate duality result known as Kakihara's theorem (see Appendix A.1 and Theorem 2 in Altamirano and Tobar (2022)). Specifically, by defining $\hat{\boldsymbol{\omega}} = \boldsymbol{\omega}_1 - \boldsymbol{\omega}_2$ and $\bar{\boldsymbol{\omega}} = (\boldsymbol{\omega}_1 + \boldsymbol{\omega}_2)/2$, the $(i,j)$-th element of matrix-valued spectral density $\mathbf{S} \triangleq [S_{ij}]_{i,j=1}^V$ is:

$$S_{ij} = \sum_{q=1}^Q w_{ij}^{(q)}\, SE_1^{(q)}(\hat{\boldsymbol{\omega}})\, SE_2^{(q)}(\bar{\boldsymbol{\omega}}),$$

where $w_{ij}^{(q)}$ are scalar weights and $SE_1^{(q)}, SE_2^{(q)}$ are squared-exponential functions (Rasmussen and Nickisch, 2010). Applying Kakihara's theorem yields the $(i,j)$-th element of corresponding kernel $k_{ij}(\mathbf{x}_1, \mathbf{x}_2) =$

$$\sum_{q=1}^Q \alpha_{ij}^{(q)} \exp\Big(-\tfrac{1}{2}(\hat{\mathbf{x}} + \boldsymbol{\theta}_{ij}^{(q)})^\top \boldsymbol{\Sigma}_{ij}^{(q)} (\hat{\mathbf{x}} + \boldsymbol{\theta}_{ij}^{(q)})\Big) \cos\Big((\hat{\mathbf{x}} + \boldsymbol{\theta}_{ij}^{(q)})^\top \boldsymbol{\mu}_{ij}^{(q)} + \phi_{ij}^{(q)}\Big) \exp\Big(-\tfrac{l_{ij}}{2}\|\bar{\mathbf{x}}\|^2\Big), \tag{2}$$

where $\hat{\mathbf{x}} = \mathbf{x}_1 - \mathbf{x}_2$, $\bar{\mathbf{x}} = (\mathbf{x}_1 + \mathbf{x}_2)/2$, and $\{\boldsymbol{\theta}_{ij}^{(q)}, \boldsymbol{\mu}_{ij}^{(q)}, \boldsymbol{\Sigma}_{ij}^{(q)}, l_{ij}\}_{q=1;i,j=1}^{Q;V}$ are the kernel hyperparameters. Equation (2) represents a *locally stationary* kernel—obtained by multiplying a stationary kernel with a non-negative modulation function—which generally limits its ability to capture strong nonstationarities (Altamirano and Tobar, 2022). To overcome this limitation, Altamirano and Tobar

**Table 1:** Simulation settings and runtime comparison for NG-SM and MOHSM

| Model | No. Iterations | No. Mixture Components ($Q$) | No. Positional Shifts ($P$) | Parameter Size | Runtimes [s] | Reference |
|---|---|---|---|---|---|---|
| MOHSM | 2000 | 2 | 2 | 29 | 63 | (Altamirano and Tobar, 2022) |
| NGSM | 2000 | 2 | N/A | 11 | 21 | (Yang et al., 2025) |

treat (2) as a base kernel $k_{ij}^p$ and localize it by replacing the global Gaussian envelope in $\bar{\mathbf{x}}$ with a window centered at a set of shift points $\{\mathbf{x}_p\}_{p=1}^P$, i.e., $k_{ij}^p(\mathbf{x}_1, \mathbf{x}_2) =$

$$\sum_{q=1}^Q \alpha_{ij}^{(q)} \exp\left(-\tfrac{1}{2}(\hat{\mathbf{x}}+\boldsymbol{\theta}_{ij}^{(q)})^\top \boldsymbol{\Sigma}_{ij}^{(q)}(\hat{\mathbf{x}}+\boldsymbol{\theta}_{ij}^{(q)})\right) \cos\left((\hat{\mathbf{x}}+\boldsymbol{\theta}_{ij}^{(q)})^\top \boldsymbol{\mu}_{ij}^{(q)}+\boldsymbol{\phi}_{ij}^{(q)}\right) \exp\left(-\tfrac{l_{ij}^{(p)}]}{2}\|\bar{\mathbf{x}}-\mathbf{x}_p\|^2\right), \quad (3)$$

The final MOHSM kernel is then the superposition of the base kernels, $k_{ij}(\mathbf{x}_1, \mathbf{x}_2) = \sum_{p=1}^P k_{ij}^p(\mathbf{x}_1, \mathbf{x}_2)$, which models nonstationarity by representing the data as a union of locally stationary regimes. Notably, the resulting spectral density exhibits a two-level mixture structure: a mixture over shift points $\mathbf{x}_p, p = 1, \ldots, P$, with each shift point further associated with a mixture of $Q$ components.

### 2.3 Revisiting the MOHSM Design

Despite targeting MOGPs, MOHSM still faces intrinsic limitations in modeling nonstationarity in contrast to NG-SM. To illustrate this limitation clearly and fairly, we consider scalar observations, under which the MOGP reduces to a standard GP, and compare the two kernels through theoretical analysis and empirical validation.

Theoretically, when $V = 1$, the two-level mixture spectral density of MOHSM is neither guaranteed to be dense (see analysis in § 3.1) in the dual space nor efficient in parameter size (Altamirano and Tobar, 2022). Consequently, MOHSM cannot approximate arbitrary nonstationary SOGP kernels, despite requiring at least $P$ times more parameters than NG-SM. In contrast, NG-SM employs a dense Gaussian mixture, enabling it to approximate any nonstationary kernel with a relatively small number of parameters.

Empirically, we compare GP regression models with MOHSM and NG-SM kernels on periodic and piecewise-periodic data with input-dependent frequency transitions (see Figure 1; simulation settings and runtimes are in Table 1). The MOHSM-based model fails to represent global input-dependent transitions, whereas the NG-SM-based model accurately models the data with significantly fewer parameters and a substantially lower runtime.

## 3 Methodology

This section begins by establishing a relaxed duality theorem between the spectral and kernel spaces, which loosens the conventional constraints on the spectral density. Building on this result, we introduce a novel nonstationary kernel for MOGPs in § 3.2. Finally, § 3.3 outlines the procedure for performing MOGP regression inference, later used in the experiments to evaluate the proposed kernel.

### 3.1 Advanced Kakihara Theorem

As discussed in § 2.3, the core limitation of the MOHSM design is due to the restrictions arising from the original Kakihara theorem (Appendix A.1). In addition, the spectral density of NG-SM kernel is constructed using a dense bivariate Gaussian mixture, which cannot be directly adopted in MOGP setting as it would also be subject to the same limitation. We now examine these restrictions in detail. For convenience, we define the measure associated with the spectral density $S_{ij}$ as $F_{ij}$.

1) $F_{ii}$ must be a positive semi-definite (PSD) measure. This requires the diagonal elements of spectral density $S_{ii}$ to be a PSD function, excluding fully flexible choices such as dense bivariate Gaussian mixtures.

2) Exchangeable within Hermitian symmetry: $F_{ij}(\boldsymbol{\omega}_1, \boldsymbol{\omega}_2) = \overline{F}_{ji}(\boldsymbol{\omega}_2, \boldsymbol{\omega}_1)$. In MOHSM, to ensure the exchangeability, the input of the spectral density is set to $\hat{\boldsymbol{\omega}} = \boldsymbol{\omega}_1 - \boldsymbol{\omega}_2$ and $\bar{\boldsymbol{\omega}} = (\boldsymbol{\omega}_1 + \boldsymbol{\omega}_2)/2$, which leads to both *reduced flexibility* and *over-parameterization*. To observe this, recall that

$S_{ij} = \prod_{p=1}^{P} \prod_{q=1}^{Q} p(\bar{\boldsymbol{\omega}}) \, p(\hat{\boldsymbol{\omega}}, \mathbf{x}_p)$, where we omit its specific functional form for brevity. First, $\hat{\boldsymbol{\omega}}$ and $\bar{\boldsymbol{\omega}}$ are modeled as independent, which removes $\hat{\boldsymbol{\omega}}$, $\bar{\boldsymbol{\omega}}$ interactions and prevents representing general bivariate spectral densities. Second, as discussed in § 2.2, over-parameterized two-level spectral mixture is required to model nonstationarity, since a single mixture layer yields only a locally stationary kernel. This local stationarity arises because, after the Fourier transform, $\bar{\boldsymbol{\omega}} = (\boldsymbol{\omega}_1 + \boldsymbol{\omega}_2)/2$ maps to $\bar{\mathbf{x}} = (\mathbf{x}_1 + \mathbf{x}_2)/2$ in the spatial domain, producing a stationary kernel, while $\hat{\boldsymbol{\omega}} = \boldsymbol{\omega}_1 - \boldsymbol{\omega}_2$ maps to $\mathbf{x}_1 - \mathbf{x}_2$, which only appears as a non-negative weighting function.

To address these constraints, we propose an advanced version of Kakihara's theorem that relaxes the above conditions, enabling more flexible choices of spectral densities.

**Theorem 2** (Advanced Kakihara Theorem). *A family of complex-valued functions $\{k_{ij}(\mathbf{x}_1, \mathbf{x}_2)\}_{i,j=1}^{V}$ on $\mathbb{R}^D$ serves as the covariance functions of a harmonizable multivariate stochastic process on $\mathbb{R}^D$ if and only if they can be represented as:*

$$k_{ij}(\mathbf{x}_1, \mathbf{x}_2) = \frac{1}{4} \iint_{\mathbb{R}^D \times \mathbb{R}^D} \Big[ \exp\big(i(\boldsymbol{\omega}_1^\top \mathbf{x}_1 - \boldsymbol{\omega}_2^\top \mathbf{x}_2)\big) + \exp\big(i(\boldsymbol{\omega}_2^\top \mathbf{x}_1 - \boldsymbol{\omega}_1^\top \mathbf{x}_2)\big)$$
$$+ \exp\big(i\boldsymbol{\omega}_1^\top (\mathbf{x}_1 - \mathbf{x}_2)\big) + \exp\big(i\boldsymbol{\omega}_2^\top (\mathbf{x}_1 - \mathbf{x}_2)\big) \Big] \, dF_{ij}(\boldsymbol{\omega}_1, \boldsymbol{\omega}_2). \qquad (4)$$

*where $\mathbf{F}(\boldsymbol{\omega}_1, \boldsymbol{\omega}_2) = [F_{ij}(\boldsymbol{\omega}_1, \boldsymbol{\omega}_2)]_{i,j=1}^{V}$ is the matrix-valued Lebesgue-Stieltjes bimeasure associated with some matrix-valued function $\mathbf{P}(\boldsymbol{\omega}_1, \boldsymbol{\omega}_2) = [P_{ij}(\boldsymbol{\omega}_1, \boldsymbol{\omega}_2)]_{i,j=1}^{V}$, which satisfies the Hermitian symmetry condition: $P_{ij}(\boldsymbol{\omega}_1, \boldsymbol{\omega}_2) = \overline{P_{ji}(\boldsymbol{\omega}_1, \boldsymbol{\omega}_2)}$.*

*Proof.* See the proof in Appendix A.1. $\qquad\square$

**Remark 1.** *We use the symbol $\mathbf{P}$ instead of $\mathbf{S}$ to denote the spectral density in order to distinguish MOHSM and our MO-LRN. Moreover, even though using the same notation, the bimeasure $\mathbf{F}$ in the two theorems differs in its functional form and associated integral representation.*

With both constraints removed, Theorem 2 implies that a Hermitian symmetric matrix-valued bivariate spectral density determines a nonstationary MOGP kernel. For flexibility, we can approximate each entry $P_{ij}$[1] using a bivariate Gaussian mixture with complex-valued coefficients, while enforcing Hermitian symmetry ($P_{ij} = \overline{P}_{ji}$). Diagonal terms are restricted to real nonnegative coefficients. Since the linear span of Gaussian atoms is dense (Plataniotis and Hatzinakos, 2000), this parameterization is, in principle, universal over admissible Hermitian symmetric spectral densities and thus can approximate any nonstationary MOGP kernel. However, realizing this flexibility requires $O(DV^2Q)$ parameters, rendering the approach impractical.

### 3.2 Multi-output low-rank nonstationary (MO-LRN) kernel

Instead of parameterizing each spectral entry with the bivariate Gaussian mixture under Hermitian symmetry, we introduce a both parameter-efficient and sufficiently expressive design of spectral density. Specifically, we first assign a latent vector $\mathbf{r}_i \in \mathbb{R}^Q$ to each output $i$ and define

$$P_{ij}(\boldsymbol{\omega}_1, \boldsymbol{\omega}_2) = \mathbf{r}_i^H \mathbf{r}_j, \qquad (5)$$

for reducing the number of parameters from $O(DV^2Q)$ to $O(DVQ)$ while automatically enforcing symmetry ($P_{ij} = \overline{P}_{ji}$). This construction closely resembles latent factor models (e.g. Koren et al., 2009; Mnih and Salakhutdinov, 2007), where pairwise interactions are captured via inner products of low-dimensional embeddings. From this perspective, the latent embedding $\mathbf{r}_i$ encodes output-specific spectral characteristics, while the relative positions of $\mathbf{r}_i$ and $\mathbf{r}_j$ in the embedding space determine the strength and sign of their cross-output spectral interactions.

To retain flexibility, we parameterize the $q$-th component in the embedding, $\mathbf{r}_i = [r_i^{(1)}, \ldots, r_i^{(Q)}]^\top$, as:

$$r_i^{(q)} = w_i^{(q)} \mathcal{N}\left( \begin{pmatrix} \boldsymbol{\omega}_1 \\ \boldsymbol{\omega}_2 \end{pmatrix} \middle| \begin{pmatrix} \boldsymbol{\mu}_{i1}^{(q)} \\ \boldsymbol{\mu}_{i2}^{(q)} \end{pmatrix}, \begin{bmatrix} \boldsymbol{\Sigma}_{i1}^{(q)} & (\boldsymbol{\Sigma}_{ic}^{(q)})^\top \\ \boldsymbol{\Sigma}_{ic}^{(q)} & \boldsymbol{\Sigma}_{i2}^{(q)} \end{bmatrix} \right), \qquad (6)$$

---

[1]For brevity, the input $(\boldsymbol{\omega}_1, \boldsymbol{\omega}_2)$ is sometimes omitted.

where $w_i^{(q)} > 0$ is the component weight, $\boldsymbol{\mu}_{i1}^{(q)}, \boldsymbol{\mu}_{i2}^{(q)} \in \mathbb{R}^D$ are the mean vectors, and the covariance blocks are diagonal matrices $\boldsymbol{\Sigma}_{i1}^{(q)} = \mathrm{diag}((\boldsymbol{\sigma}_{i1}^{(q)})^2)$, $\boldsymbol{\Sigma}_{i2}^{(q)} = \mathrm{diag}((\boldsymbol{\sigma}_{i2}^{(q)})^2)$, with $\boldsymbol{\sigma}_{i1}^{(q)}, \boldsymbol{\sigma}_{i2}^{(q)} \in \mathbb{R}^D$. The cross-covariance is defined as $\boldsymbol{\Sigma}_{ic}^{(q)} = \rho_i^{(q)} \mathrm{diag}(\boldsymbol{\sigma}_{i1}^{(q)}) \mathrm{diag}(\boldsymbol{\sigma}_{i2}^{(q)})$, where $\rho_i^{(q)} \in [-1, 1]$ denotes the correlation coefficient.

Substituting (6) into (5) yields:

$$P_{ij}(\boldsymbol{\omega}_1, \boldsymbol{\omega}_2) = \mathbf{r}_i^H \mathbf{r}_j = \sum_{q=1}^{Q} z_{ij}^{(q)} \mathcal{N}\left( \begin{pmatrix} \boldsymbol{\omega}_1 \\ \boldsymbol{\omega}_2 \end{pmatrix} \middle| \boldsymbol{m}_{ij}^{(q)}, \mathbf{S}_{ij}^{(q)} \right), \tag{7}$$

which remains a bivariate Gaussian mixture. Here, $z_{ij}^{(q)}$, $\boldsymbol{m}_{ij}^{(q)} \in \mathbb{R}^{2D}$ and $\mathbf{S}_{ij}^{(q)} \in \mathbb{R}^{2D \times 2D}$ are the scaling factor, mean vector, and covariance matrix of the $q$-the bivariate Gaussian density. These quantities are entirely determined by the paired component parameters $\boldsymbol{\theta} = \{w_k^{(q)}, \boldsymbol{\mu}_{k1}^{(q)}, \boldsymbol{\mu}_{k2}^{(q)}, \boldsymbol{\sigma}_{k1}^{(q)}, \boldsymbol{\sigma}_{k2}^{(q)}, \rho_k^{(q)}\}_{k \in \{i,j\}}$. Further details on the parameter derivations are given in Appendix A.2.

To ensure the resulting kernel is real-valued, we set each $P_{ij}(\boldsymbol{\omega}_1, \boldsymbol{\omega}_2) = \frac{1}{2}\big[P_{ij}(\boldsymbol{\omega}_1, \boldsymbol{\omega}_2) + P_{ij}(-\boldsymbol{\omega}_1, -\boldsymbol{\omega}_2)\big]$. By applying the duality relation in Eq. (4) of Theorem 2, we obtain a real-valued MOGP kernel, whose explicit form is given below.

**Definition 1** (Multi-output Low-Rank Nonstationary Kernel)**.** *The $(i,j)$-th element of the MO-LRN kernel $k_{ij}(\mathbf{x}_1, \mathbf{x}_2)$ is defined as*

$$= \sum_{q=1}^{Q} \frac{z_{ij}^{(q)}}{4} \Bigg[ \cos\left((\boldsymbol{m}_{ij1}^{(q)})^\top \mathbf{x}_1 - (\boldsymbol{m}_{ij2}^{(q)})^\top \mathbf{x}_2\right) \exp\left(-\tfrac{1}{2}(\mathbf{x}_1^\top \boldsymbol{S}_{ij1}^{(q)} \mathbf{x}_1 - 2\mathbf{x}_1^\top (\boldsymbol{S}_{ijc}^{(q)})^\top \mathbf{x}_2 + \mathbf{x}_2^\top \boldsymbol{S}_{ij2}^{(q)} \mathbf{x}_2)\right)$$

$$+ \cos\left((\boldsymbol{m}_{ij2}^{(q)})^\top \mathbf{x}_1 - (\boldsymbol{m}_{ij1}^{(q)})^\top \mathbf{x}_2\right) \exp\left(-\tfrac{1}{2}(\mathbf{x}_2^\top \boldsymbol{S}_{ij1}^{(q)} \mathbf{x}_2 - 2\mathbf{x}_1^\top \boldsymbol{S}_{ijc}^{(q)} \mathbf{x}_2 + \mathbf{x}_1^\top \boldsymbol{S}_{ij2}^{(q)} \mathbf{x}_1)\right)$$

$$+ \cos\left((\boldsymbol{m}_{ij1}^{(q)})^\top (\mathbf{x}_1 - \mathbf{x}_2)\right) \exp\left(-\tfrac{1}{2}(\mathbf{x}_1 - \mathbf{x}_2)^\top \boldsymbol{S}_{ij1}^{(q)} (\mathbf{x}_1 - \mathbf{x}_2)\right)$$

$$+ \cos\left((\boldsymbol{m}_{ij2}^{(q)})^\top (\mathbf{x}_1 - \mathbf{x}_2)\right) \exp\left(-\tfrac{1}{2}(\mathbf{x}_1 - \mathbf{x}_2)^\top \boldsymbol{S}_{ij2}^{(q)} (\mathbf{x}_1 - \mathbf{x}_2)\right) \Bigg]$$

*where each component has parameters $\big(z_{ij}^{(q)}, \boldsymbol{m}_{ij}^{(q)}, \boldsymbol{S}_{ij}^{(q)}\big)$ with partitions $\boldsymbol{m}_{ij}^{(q)} = [\boldsymbol{m}_{ij1}^{(q)\top}, \boldsymbol{m}_{ij2}^{(q)\top}]^\top$ and $\boldsymbol{S}_{ij}^{(q)} = \begin{bmatrix} \boldsymbol{S}_{ij1}^{(q)} & (\boldsymbol{S}_{ijc}^{(q)})^\top \\ \boldsymbol{S}_{ijc}^{(q)} & \boldsymbol{S}_{ij2}^{(q)} \end{bmatrix}$, all analytically determined from hyperparameters $\boldsymbol{\theta}$, with the explicit derivation steps given in Definition 2 in Appendix A.2.*

**Remark 2.** *Although each element of the spectral density $P_{ij}$ appears to be modeled as a bivariate Gaussian mixture, denseness can only be guaranteed for the diagonal terms ($i = j$), since only these are independently parameterized. Nevertheless, this construction remains more expressive than MOHSM, whose diagonal spectral terms are not even guaranteed to be dense, as discussed in § 2.3.*

### 3.3 MULTI-OUTPUT GAUSSIAN PROCESS REGRESSION

The training procedure for MOGP regression parallels that of the SOGP–by maximizing the marginal log-likelihood of the observed data with respect to the hyperparameters. Formally, we represent each data point as a pair $\{(\mathbf{x}_n, \mathbf{y}_n)\}_{n=1}^N$, where $\mathbf{x}_n \in \mathbb{R}^D$ and $\mathbf{y}_n \in \mathbb{R}^V$. Let $\mathbf{y} = [\mathbf{y}_1^\top, \ldots, \mathbf{y}_N^\top]^\top \in \mathbb{R}^{NV}$ denote the vertically stacked outputs.

We assume that the observations $\mathbf{y}$ are generated from the inputs $\mathbf{X}$ via a noisy MOGP mapping. Specifically,

$$\mathbf{y} = \mathrm{vec}(\boldsymbol{\mathcal{F}}) + \boldsymbol{\epsilon}, \quad \boldsymbol{\epsilon} \sim \mathcal{N}(0, \boldsymbol{\Sigma}_n), \quad \mathrm{vec}(\boldsymbol{\mathcal{F}}) \sim \mathcal{N}(0, \mathcal{K}(\mathbf{X}, \mathbf{X})), \tag{8}$$

where $\mathrm{vec}(\boldsymbol{\mathcal{F}})$ denote the vectorized latent function evaluations over all outputs and inputs, $\mathcal{K}(\mathbf{X}, \mathbf{X})$ is the MOGP prior covariance matrix, and $\boldsymbol{\Sigma}_n$ is the observation noise covariance, taking the form $\boldsymbol{\Sigma}_n = \mathbf{I}_N \otimes \mathrm{diag}(\sigma_1^2, \ldots, \sigma_V^2)$, with $\sigma_i^2 > 0$ is the noise variance specific to the $i$-th output.

Due to the conjugacy of the Gaussian prior and Gaussian likelihood, we can marginalize out $\mathrm{vec}(\boldsymbol{\mathcal{F}})$ to obtain the marginal log-likelihood:

$$\log p(\mathbf{y} \mid \boldsymbol{\theta}) = -\frac{1}{2}\mathbf{y}^\top (\mathcal{K} + \boldsymbol{\Sigma}_n)^{-1} \mathbf{y} - \frac{1}{2}\log|\mathcal{K} + \boldsymbol{\Sigma}_n| - \frac{N}{2}\log 2\pi, \tag{9}$$

Maximizing $\log p(\mathbf{y} \mid \boldsymbol{\theta})$ yields the maximum likelihood estimates of hyperparameters $\boldsymbol{\theta}$.

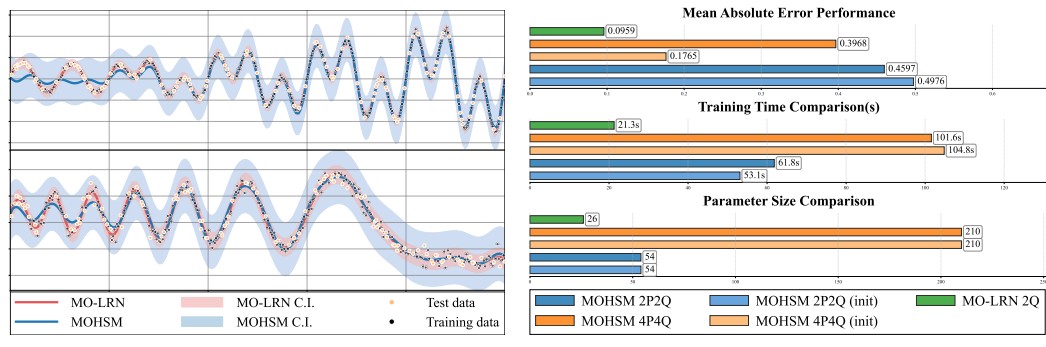

**(a)** MOGP regression plot        **(b)** Comparison on model performance

**Figure 2:** MOGP regression results comparing MO-LRN and MOHSM on synthetic data. Panel (a) shows regression plots, while Panel (b) reports MAE, training time, and parameter size.

**Table 2:** Comparison of various MOGP kernels. An expressive kernel refers to one whose spectral density is flexible in the dual space. Here, $I$ denotes the index in the LMC summation over latent processes, with $I \leq V$.

| Model | Able to model non-stationary pattern | Able to model stationary pattern | Expressive kernel | #Parameters | Reference |
|---|---|---|---|---|---|
| MOSM | ✗ | ✓ | ✓ | $O(DQV)$ | (Parra and Tobar, 2017) |
| CONV | ✗ | ✓ | ✗ | $O(DQV)$ | (Alvarez and Lawrence, 2008) |
| LMC-SM | ✗ | ✓ | ✗ | $O(IDQ)$ | (Wilson and Adams, 2013a) |
| LMC-NGSM | ✓ | ✓ | ✗ | $O(IDQ)$ | (Yang et al., 2025) |
| MOHSM | ✓ | ✓ | ✗ | $O(PDQV)$ | (Altamirano and Tobar, 2022) |
| MO-LRN | ✓ | ✓ | ✓ | $O(DQV)$ | This Work |

# 4 EXPERIMENTAL RESULTS

This section first presents a detailed comparison between our MO-LRN kernel and the MOHSM kernel on MOGP regression using synthetic datasets (§ 4.1). We then show that MO-LRN achieves superior performance on real-world MOGP regression tasks (§ 4.2), as well as on missing-data interpolation and imputation tasks (§ 4.3) across diverse time-series datasets. Comprehensive experimental setups are provided in Appendix B, and benchmark implementations in Appendix B.2 and B.3. Appendix C.2 reports additional regression experiments on other real-world datasets.

## 4.1 SYNTHETIC EXAMPLE

In this subsection, we conduct a detailed comparison between the proposed MO-LRN kernel and the MOHSM kernel on a MOGP regression task. To this end, we construct a two-dimensional output $\mathbf{y}_n$ by combining the scalar periodic signal and the piecewise-periodic signal introduced in §2.3 (see more detail in Appendix B.1.1). Collecting $\{\mathbf{y}_n\}_{n=1}^N$ forms the complete dataset, on which we perform MOGP regression with both kernels.

For MOHSM, performance is highly sensitive to the choice of $(P, Q)$ and requires carefully designed initialization strategies (Altamirano and Tobar, 2022). We therefore compare our kernel with MOHSM across multiple settings, evaluating three metrics: test set accuracy measured by the mean absolute error (MAE), training time, and parameter count. As shown in Figure 2b, our kernel consistently outperforms MOHSM while requiring substantially fewer parameters and a significantly shorter training time. By contrast, MOHSM not only delivers inferior accuracy but also remains dependent on costly initialization and a large parameter budget.

To visualize the performance gap between the two kernels, Figure 2a shows the regression fits obtained using our kernel and the MOHSM kernel under its lowest-MAE configuration ($P = 4, Q = 4$ with the initialization strategy). The results clearly indicate that the MOGP model with our kernel can precisely fit the data and effectively capture nonstationary patterns. In contrast, although the MOHSM-based model outperforms a baseline that applies SOGP regression independently to each dimension (as shown in Figure 1) by leveraging cross-output correlations, it still fails to fully capture nonstationary patterns due to its limited spectral density design, thus resulting in inferior overall performance.

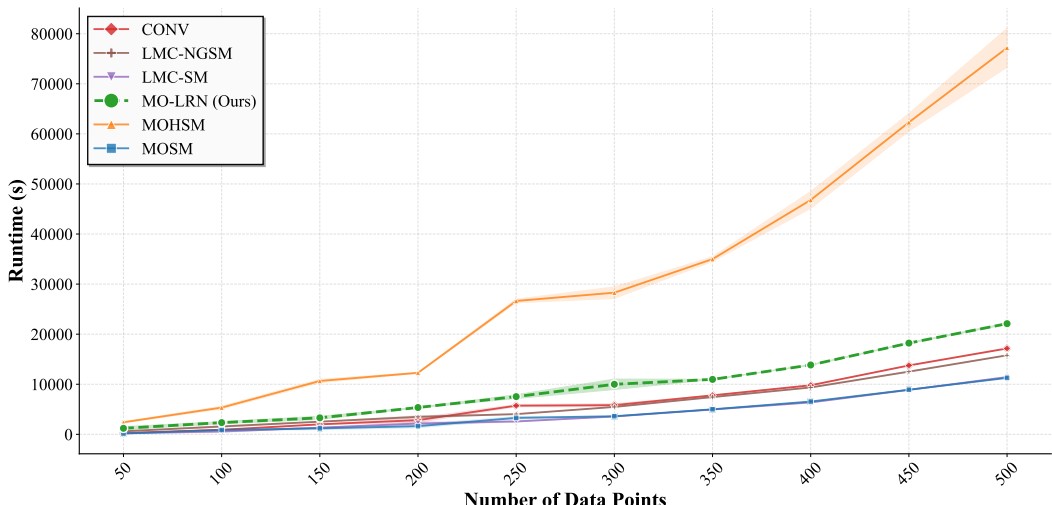

**Figure 3:** Runtime comparison on the ETT dataset against a varying number of data points.

**Table 3:** Comparison of MAE, NMAE, RMSE and NLPD across kernels on the ETT dataset. Each column (OT, HUFL, HULL, MUFL, MULL, LUFL, LULL) reports the reconstruction accuracy for one output variable, while the last column (Overall) summarizes the average performance across all outputs. Mean and standard deviation are computed over five runs. The best result per column is bolded, and the second-best is underlined.

| TARGET | OT | HUFL | HULL | MUFL | MULL | LUFL | LULL | OVERALL |
|---|---|---|---|---|---|---|---|---|
| METRIC | | | | MAE | | | | |
| CONV | $0.276 \pm 0.053$ | $0.439 \pm 0.140$ | $0.214 \pm 0.047$ | $0.455 \pm 0.136$ | $0.781 \pm 0.119$ | $0.348 \pm 0.091$ | $0.220 \pm 0.039$ | $0.390 \pm 0.073$ |
| LMC-SM | $0.274 \pm 0.007$ | $0.329 \pm 0.003$ | $0.223 \pm 0.002$ | $0.357 \pm 0.004$ | $0.649 \pm 0.048$ | $0.410 \pm 0.004$ | $0.306 \pm 0.037$ | $0.364 \pm 0.014$ |
| MOHSM | $0.289 \pm 0.005$ | $0.328 \pm 0.003$ | $0.251 \pm 0.006$ | $0.365 \pm 0.004$ | $0.389 \pm 0.007$ | $0.359 \pm 0.002$ | $0.290 \pm 0.001$ | $0.325 \pm 0.001$ |
| MOSM | $0.295 \pm 0.024$ | $0.309 \pm 0.036$ | $0.216 \pm 0.027$ | $0.333 \pm 0.039$ | $0.459 \pm 0.017$ | $0.379 \pm 0.022$ | $0.210 \pm 0.016$ | $0.314 \pm 0.014$ |
| LMC-NGSM | $0.204 \pm 0.018$ | $\underline{0.271 \pm 0.007}$ | $\underline{0.167 \pm 0.010}$ | $0.293 \pm 0.002$ | $\underline{0.303 \pm 0.033}$ | $0.372 \pm 0.014$ | $\underline{0.180 \pm 0.002}$ | $\underline{0.256 \pm 0.011}$ |
| **MO-LRN** | $\mathbf{0.148 \pm 0.003}$ | $\mathbf{0.256 \pm 0.017}$ | $\mathbf{0.125 \pm 0.005}$ | $\mathbf{0.287 \pm 0.023}$ | $\mathbf{0.208 \pm 0.008}$ | $\mathbf{0.248 \pm 0.003}$ | $\mathbf{0.133 \pm 0.003}$ | $\mathbf{0.201 \pm 0.006}$ |
| METRIC | | | | NMAE | | | | |
| CONV | $0.294 \pm 0.038$ | $0.424 \pm 0.131$ | $0.229 \pm 0.033$ | $0.457 \pm 0.118$ | $0.689 \pm 0.335$ | $0.468 \pm 0.090$ | $0.303 \pm 0.119$ | $0.409 \pm 0.058$ |
| LMC-SM | $0.319 \pm 0.008$ | $0.369 \pm 0.003$ | $0.266 \pm 0.002$ | $0.409 \pm 0.005$ | $0.771 \pm 0.057$ | $0.488 \pm 0.005$ | $0.331 \pm 0.040$ | $0.422 \pm 0.016$ |
| MOHSM | $0.337 \pm 0.006$ | $0.367 \pm 0.003$ | $0.299 \pm 0.008$ | $0.418 \pm 0.004$ | $0.463 \pm 0.008$ | $0.428 \pm 0.002$ | $0.313 \pm 0.001$ | $0.375 \pm 0.001$ |
| MOSM | $0.344 \pm 0.028$ | $0.346 \pm 0.041$ | $0.258 \pm 0.033$ | $0.382 \pm 0.044$ | $0.546 \pm 0.020$ | $0.452 \pm 0.026$ | $0.227 \pm 0.017$ | $0.365 \pm 0.016$ |
| LMC-NGSM | $0.238 \pm 0.021$ | $\underline{0.303 \pm 0.008}$ | $\underline{0.199 \pm 0.012}$ | $0.336 \pm 0.002$ | $\underline{0.361 \pm 0.039}$ | $0.444 \pm 0.016$ | $\underline{0.194 \pm 0.002}$ | $\underline{0.296 \pm 0.013}$ |
| **MO-LRN** | $\mathbf{0.172 \pm 0.003}$ | $\mathbf{0.287 \pm 0.019}$ | $\mathbf{0.149 \pm 0.006}$ | $\mathbf{0.328 \pm 0.027}$ | $\mathbf{0.247 \pm 0.009}$ | $\mathbf{0.296 \pm 0.004}$ | $\mathbf{0.144 \pm 0.003}$ | $\mathbf{0.232 \pm 0.007}$ |
| METRIC | | | | RMSE | | | | |
| CONV | $0.317 \pm 0.018$ | $0.453 \pm 0.107$ | $0.252 \pm 0.014$ | $0.490 \pm 0.091$ | $0.754 \pm 0.282$ | $0.558 \pm 0.169$ | $0.346 \pm 0.151$ | $0.453 \pm 0.020$ |
| LMC-SM | $0.340 \pm 0.030$ | $0.389 \pm 0.035$ | $0.280 \pm 0.033$ | $0.439 \pm 0.045$ | $0.796 \pm 0.079$ | $0.549 \pm 0.075$ | $0.376 \pm 0.048$ | $0.453 \pm 0.045$ |
| MOHSM | $0.376 \pm 0.005$ | $0.407 \pm 0.004$ | $0.333 \pm 0.005$ | $0.467 \pm 0.002$ | $0.574 \pm 0.005$ | $0.540 \pm 0.001$ | $0.379 \pm 0.001$ | $0.440 \pm 0.002$ |
| MOSM | $0.386 \pm 0.027$ | $0.387 \pm 0.047$ | $0.298 \pm 0.032$ | $0.436 \pm 0.049$ | $0.679 \pm 0.028$ | $0.563 \pm 0.026$ | $0.269 \pm 0.020$ | $0.431 \pm 0.017$ |
| LMC-NGSM | $\underline{0.261 \pm 0.019}$ | $\underline{0.344 \pm 0.009}$ | $\underline{0.226 \pm 0.011}$ | $\mathbf{0.392 \pm 0.003}$ | $\underline{0.469 \pm 0.034}$ | $0.522 \pm 0.023$ | $\underline{0.234 \pm 0.001}$ | $\underline{0.350 \pm 0.013}$ |
| **MO-LRN** | $\mathbf{0.210 \pm 0.003}$ | $\mathbf{0.334 \pm 0.023}$ | $\mathbf{0.188 \pm 0.004}$ | $\underline{0.393 \pm 0.030}$ | $\mathbf{0.412 \pm 0.007}$ | $\mathbf{0.353 \pm 0.003}$ | $\mathbf{0.178 \pm 0.002}$ | $\mathbf{0.295 \pm 0.008}$ |
| METRIC | | | | NLPD | | | | |
| CONV | $0.384 \pm 0.045$ | $0.484 \pm 0.034$ | $0.156 \pm 0.060$ | $0.647 \pm 0.147$ | $1.328 \pm 0.023$ | $0.877 \pm 0.069$ | $0.748 \pm 0.320$ | $0.661 \pm 0.069$ |
| LMC-SM | $0.286 \pm 0.021$ | $0.429 \pm 0.009$ | $0.066 \pm 0.012$ | $0.554 \pm 0.008$ | $1.283 \pm 0.033$ | $0.889 \pm 0.002$ | $0.506 \pm 0.198$ | $0.573 \pm 0.031$ |
| MOHSM | $0.733 \pm 0.022$ | $0.757 \pm 0.019$ | $0.702 \pm 0.023$ | $0.809 \pm 0.016$ | $0.920 \pm 0.011$ | $0.881 \pm 0.011$ | $0.740 \pm 0.021$ | $0.792 \pm 0.018$ |
| MOSM | $0.473 \pm 0.073$ | $0.467 \pm 0.118$ | $0.208 \pm 0.118$ | $0.586 \pm 0.116$ | $1.048 \pm 0.038$ | $0.846 \pm 0.047$ | $\underline{0.103 \pm 0.075}$ | $0.533 \pm 0.043$ |
| LMC-NGSM | $\underline{0.146 \pm 0.167}$ | $\underline{0.365 \pm 0.046}$ | $\underline{-0.019 \pm 0.133}$ | $\mathbf{0.501 \pm 0.020}$ | $\underline{0.866 \pm 0.367}$ | $0.771 \pm 0.049$ | $0.042 \pm 0.172$ | $\underline{0.382 \pm 0.133}$ |
| **MO-LRN** | $\mathbf{-0.159 \pm 0.012}$ | $\mathbf{0.334 \pm 0.087}$ | $\mathbf{-0.271 \pm 0.038}$ | $\underline{0.505 \pm 0.097}$ | $\mathbf{0.691 \pm 0.092}$ | $\mathbf{0.385 \pm 0.007}$ | $\mathbf{-0.323 \pm 0.014}$ | $\mathbf{0.166 \pm 0.031}$ |

## 4.2 ELECTRICAL TRANSFORMER TEMPERATURE DATA

We further evaluate our kernel on a MOGP regression task using the real-world electricity transformer temperature (ETT) dataset (Zhou et al., 2021), which contains one-week records of oil temperature and six load-related features We treat time as the input and the oil temperature together with the six load-related features as the observation vector[2]. The dataset is randomly split into 70% for training and 30% for testing.

For fair comparison, we benchmark against both stationary and nonstationary kernels. The stationary baselines include: (i) the multi-output spectral mixture (MOSM) kernel (Parra and Tobar, 2017), (ii) the convolution (CONV) kernel (Alvarez and Lawrence, 2008), and (iii) the LMC with a spectral

---

[2]See Appendix B.1.2 for a detailed description of the dataset and preprocessing.

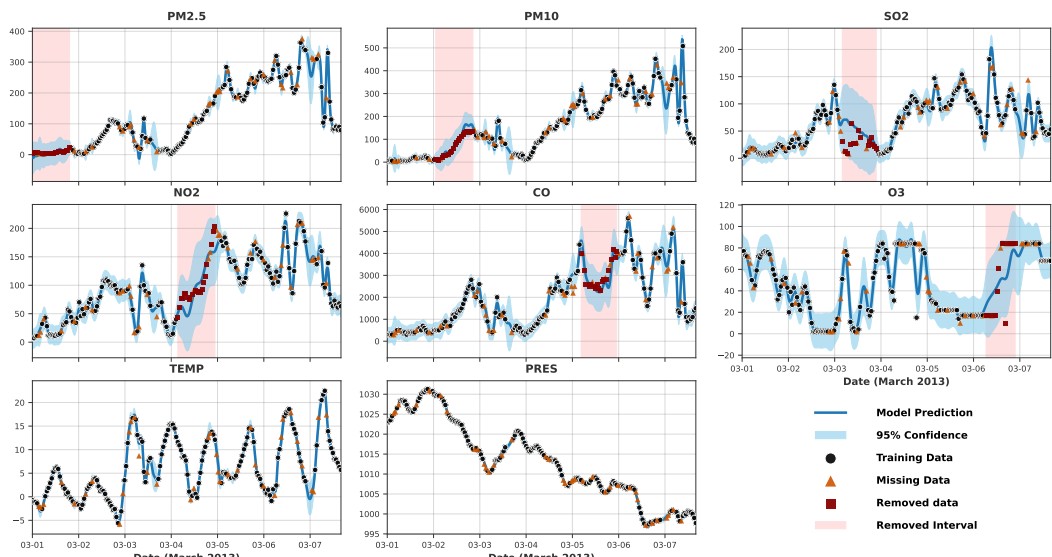

**Figure 4:** Interpolation and Imputation plots on air quality dataset

mixture base kernel (LMC-SM) (Álvarez et al., 2010; Wilson and Adams, 2013b). The nonstationary baselines include: (i) the LMC with a NG-SM base kernel (LMC-NGSM) (Yang et al., 2025; Álvarez et al., 2010), and (ii) the MOHSM kernel (Altamirano and Tobar, 2022)[3]. More detailed comparison can be found in Table 2.

Each kernel is evaluated on the test set over five independent runs, and Table 3 reports the mean and standard deviation of MAE, normalized MAE (NMAE), RMSE, and negative log predictive density (NLPD). It can be seen that our proposed MO-LRN kernel outperforms all others across the four metrics, while LMC-NGSM almost achieves the second-best performance. This implies that the ETT dataset naturally contains nonstationary patterns. Consequently, stationary kernels such as MOSM, LMC-SM, and CONV exhibit inferior performance compared to LMC-NGSM, despite it being a relatively simple nonstationary MOGP kernel.

For nonstationary kernels, both LMC-NGSM and MOHSM perform worse than our proposed kernel, as they are theoretically limited in expressiveness from a spectral perspective. From the viewpoint of spectral density design, MOHSM should, in principle, outperform LMC-NGSM kernel. However, this is not observed in practice, likely because its large number of parameters makes training substantially more challenging, potentially leading to convergence to a suboptimal solution and preventing it from realizing its theoretical advantages.

Figure 3 reports the runtime of the benchmark MOGP kernels on the ETT dataset as the number of data points increases. Our proposed MO-LRN kernel exhibits computational efficiency comparable to existing kernels, with the exception of the MOHSM kernel, whose runtime grows rapidly and quickly approaches the upper limit.

### 4.3 Air Quality Data

In this subsection, we evaluate kernel robustness to missing data in two tasks: imputation (reconstructing large continuous missing segments) and interpolation (predicting at isolated held-out points). For this purpose, we use data from the Aotizhongxin station of the Beijing multi-site air-quality dataset (Zhang et al., 2017) for March 1–8, 2013, comprising 184 hourly records of eight outputs: six pollutants (PM2.5, PM10, SO2, NO2, CO, O3) and two meteorological variables (TEMP, PRES), with no original missing values (see more detail in Appendix B.1.3). For imputation, we remove different continuous intervals from each pollutant output; for interpolation, we randomly drop 20% of isolated points from all outputs. We then train an MOGP regression model on the resulting datasets containing both types of missing data.

---

[3]See Appendix B.2 for benchmark implementation details.

**Table 4:** MAE for interpolation and imputation tasks across different kernels on the air quality dataset. Each column (PM2.5, PM10, SO2, NO2, CO, O3, TEMP, PRES) corresponds to one target variable, and the last column (Overall) summarizes the average performance across all targets. Mean and standard deviation are computed over 5 runs. The best (lowest) values are highlighted in bold, and the second-best are underlined.

| TARGET | PM2.5 | | PM10 | | SO2 | | NO2 | |
|---|---|---|---|---|---|---|---|---|
| | Interp. | Imput. | Interp. | Imput. | Interp. | Imput. | Interp. | Imput. |
| MOHSM | $0.237 \pm 0.024$ | $0.411 \pm 0.108$ | $0.244 \pm 0.036$ | $0.331 \pm 0.103$ | $0.449 \pm 0.020$ | $\mathbf{0.375 \pm 0.116}$ | $0.315 \pm 0.056$ | $0.345 \pm 0.153$ |
| CONV | $0.156 \pm 0.041$ | $0.148 \pm 0.102$ | $0.173 \pm 0.036$ | $0.077 \pm 0.019$ | $0.401 \pm 0.160$ | $0.448 \pm 0.093$ | $0.332 \pm 0.082$ | $0.448 \pm 0.073$ |
| LMC-SM | $0.116 \pm 0.006$ | $\mathbf{0.062 \pm 0.030}$ | $0.152 \pm 0.004$ | $\mathbf{0.051 \pm 0.003}$ | $0.381 \pm 0.109$ | $0.684 \pm 0.299$ | $0.317 \pm 0.077$ | $0.796 \pm 0.354$ |
| MOSM | $0.241 \pm 0.041$ | $0.486 \pm 0.357$ | $0.213 \pm 0.051$ | $0.161 \pm 0.097$ | $0.393 \pm 0.103$ | $0.619 \pm 0.180$ | $0.285 \pm 0.116$ | $0.434 \pm 0.123$ |
| LMC-NGSM | $\mathbf{0.113 \pm 0.008}$ | $0.093 \pm 0.006$ | $\mathbf{0.136 \pm 0.001}$ | $0.051 \pm 0.002$ | $0.468 \pm 0.130$ | $0.462 \pm 0.116$ | $0.397 \pm 0.008$ | $0.486 \pm 0.022$ |
| MO-LRN | $0.122 \pm 0.019$ | $0.109 \pm 0.079$ | $0.137 \pm 0.012$ | $0.103 \pm 0.016$ | $\mathbf{0.273 \pm 0.035}$ | $0.484 \pm 0.115$ | $\mathbf{0.187 \pm 0.028}$ | $\mathbf{0.325 \pm 0.041}$ |

| TARGET | CO | | O3 | | TEMP | PRES | Overall | |
|---|---|---|---|---|---|---|---|---|
| | Interp. | Imput. | Interp. | Imput. | Interp. | Interp. | Interp. | Imput. |
| MOHSM | $0.310 \pm 0.022$ | $0.465 \pm 0.048$ | $0.385 \pm 0.008$ | $0.982 \pm 0.065$ | $0.347 \pm 0.015$ | $0.118 \pm 0.024$ | $0.301 \pm 0.011$ | $0.485 \pm 0.036$ |
| CONV | $0.328 \pm 0.044$ | $0.526 \pm 0.111$ | $0.421 \pm 0.009$ | $\mathbf{0.674 \pm 0.015}$ | $0.676 \pm 0.297$ | $0.041 \pm 0.049$ | $0.316 \pm 0.042$ | $0.387 \pm 0.043$ |
| LMC-SM | $0.275 \pm 0.019$ | $0.196 \pm 0.035$ | $0.483 \pm 0.170$ | $1.188 \pm 0.052$ | $0.135 \pm 0.010$ | $0.507 \pm 0.446$ | $0.296 \pm 0.038$ | $0.496 \pm 0.102$ |
| MOSM | $0.314 \pm 0.018$ | $0.497 \pm 0.153$ | $\mathbf{0.318 \pm 0.040}$ | $0.714 \pm 0.356$ | $0.450 \pm 0.211$ | $0.098 \pm 0.022$ | $0.289 \pm 0.024$ | $0.485 \pm 0.063$ |
| LMC-NGSM | $0.278 \pm 0.009$ | $\mathbf{0.175 \pm 0.006}$ | $0.449 \pm 0.186$ | $0.965 \pm 0.138$ | $0.137 \pm 0.007$ | $\mathbf{0.020 \pm 0.003}$ | $0.250 \pm 0.007$ | $0.372 \pm 0.037$ |
| MO-LRN | $\mathbf{0.220 \pm 0.034}$ | $0.360 \pm 0.118$ | $0.328 \pm 0.016$ | $0.723 \pm 0.054$ | $\mathbf{0.132 \pm 0.016}$ | $0.022 \pm 0.005$ | $\mathbf{0.178 \pm 0.008}$ | $\mathbf{0.351 \pm 0.032}$ |

Figure 4 presents the results of imputation and interpolation with associated confidence intervals using an MOGP model equipped with our MO-LRN kernel. For interpolation, the model accurately estimates isolated missing values (orange triangles) with narrow confidence bounds. For imputation, the reconstruction of large continuous missing segments (red squares) yields broad but accurate confidence intervals that fully cover the ground-truth values. Overall, these results indicate that the MO-LRN kernel effectively captures both intra- and cross-output correlations.

Table 4 compares the MAE for interpolation and imputation across different kernels, averaged over five runs. The model with MO-LRN attains the lowest errors in both tasks, excelling in reconstructing large continuous missing segments and isolated missing values. Stationary kernels—MOSM, LMC-SM, and CONV—cannot adapt to input-dependent patterns, resulting in weaker performance. Among non-stationary kernels, LMC-NGSM lacks the capacity to capture cross-output correlations, and MOHSM is constrained by its spectral design, leading to inferior performance in both tasks compared with MO-LRN.

## 5 CONCLUSIONS

In this paper, we introduce the MO-LRN kernel, a novel nonstationary MOGP kernel grounded in a new spectral–kernel duality that eliminates restrictive structural constraints and, in principle, permits fully flexible matrix-valued spectral densities. To avoid the quadratic parameter growth of this unconstrained form, MO-LRN adopts a low-rank spectral density with independent bivariate Gaussian-mixture factors, reducing complexity to linear in the number of outputs while preserving sufficient modeling expressiveness. Extensive experiments on synthetic and real-world datasets for regression, interpolation, and missing-data imputation tasks demonstrate that MO-LRN consistently outperforms existing MOGP kernels.

## 6 ACKNOWLEDGMENTS

Michael Zhang was partially supported by the University of Hong Kong Seed Fund for PI Research #2402101367.

Pablo M. Olmos has received funding from the EU Horizon Europe research and innovationprogram under the Marie Skłodowska-Curie Grant agreement No. 101226456: MLCARE, Machine Learning Computational Advancements for peRsonalized mEdicine. Views and opinions expressed are however those of the author(s) only and do not necessarily reflect those of the EU or the European Commission (EC). Neither the EU nor the EC can be held responsible for them.

Pablo M. Olmos was also supported by the Comunidad de Madrid IND2024/TIC-34728, IDEA-CM project (TEC-2024/COM-89), the ELLIS Unit Madrid (European Laboratory for Learning and Intelligent Systems), the 2024 Leonardo Grant for Scientific Research and Cultural Creation from the BBVA Foundation, and from the Ministerio de Ciencia, Innovacion y Universidades, Spain, under Grant Agreements No. PID2024- 157856NB-I00 CARTESIAN,PID2021-123182OB-I00 EPiCENTER (funded by MICIU/AEI/ 10.13039/501100011033 and by ERDF/UE).

**Ethics Statement:** This work is primarily theoretical, introducing the multi-output low-rank non-stationary (MO-LRN) kernel for multi-output Gaussian processes. In the spirit of scientific integrity and transparency, the details of all experiments conducted and the open access datasets used are thoroughly described in the paper. Despite its theoretical focus, we recognize that any powerful predictive tool, once deployed, carries potential risks. A key concern is the potential for a model using our kernel to generate harmful or misleading information, such as perpetuating societal biases if trained on flawed data. The responsibility thus lies with the practitioner to audit their data and deployment context, ensuring the application is vetted for fairness and does not lead to negative societal consequences.

**Reproducibility Statement:** We provide the source code for a toy example with corresponding output in the supplementary materials to demonstrate the core functionality of our proposed method. To replicate our main experimental results, the Appendix B contains all necessary details, including the specific hyperparameters, descriptions of the datasets, and our data processing pipeline. We commit to releasing the full, documented codebase on a public repository upon acceptance of the paper to ensure complete verification and to support future research.

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

APPENDIX

# A  ADVANCED KAKIHARA THEOREM AND MULTI-OUTPUT LOW-RANK NONTATIONARY KERNEL

## A.1  PROOF OF THEOREM 2

Before proving Theorem 2, we first recall Kakihara's Theorem (Kakihara, 1997) and introduce a lemma that reformulates its conditions on the measure $F_{ij}$ in terms of the spectral density.

**Theorem 3** (Kakihara's Theorem (Kakihara, 1997)). *A family of complex-valued covariance functions $\{k_{ij}(\mathbf{x}_1, \mathbf{x}_2)\}_{i,j=1}^{V}$ on $\mathbb{R}^D$ corresponds to a harmonizable multivariate stochastic process if and only if each function admits the representation*

$$k_{ij}(\mathbf{x}_1, \mathbf{x}_2) = \iint_{\mathbb{R}^D \times \mathbb{R}^D} e^{i(\boldsymbol{\omega}_1^\top \mathbf{x}_1 - \boldsymbol{\omega}_2^\top \mathbf{x}_2)} \, dF_{ij}(\boldsymbol{\omega}_1, \boldsymbol{\omega}_2), \tag{10}$$

*where $\mathbf{F} = \{F_{ij}(A, B)\}_{i,j=1}^{V}$ is a matrix-valued spectral bimeasure such that:*

*(1) For all $i$, $F_{ii}$ is a positive semi-definite (PSD) measure;*

*(2) For all $i, j$, $F_{ij}(A, B) = \overline{F_{ji}(B, A)}$ for all measurable sets $A, B \subseteq \mathbb{R}^D$.*

**Lemma 1.** *Let $F_{ij}(A, B) = \int_A \int_B S_{ij}(\boldsymbol{\omega}_1, \boldsymbol{\omega}_2) \, d\boldsymbol{\omega}_2 \, d\boldsymbol{\omega}_1$, where $S_{ij}$ is the spectral density. The two conditions of Kakihara's theorem, namely*

*(1) for all $i$, $F_{ii}$ is a PSD measure;*

*(2) for all $i, j$ and measurable $A, B \subseteq \mathbb{R}^D$, $F_{ij}(A, B) = \overline{F_{ji}(B, A)}$,*

*hold if and only if the spectral matrix $S$ satisfies:*

*A) $S_{ii}$ is PSD for each diagonal entry;*

*B) $S_{ij}(\boldsymbol{\omega}_1, \boldsymbol{\omega}_2) = \overline{S_{ji}(\boldsymbol{\omega}_2, \boldsymbol{\omega}_1)}$ for all $i, j$.*

Now, we prove Theorem 2 by showing necessity ($\Rightarrow$) and sufficiency ($\Leftarrow$) as below.

**Necessity ($\Rightarrow$):** From Valid Kernel to the Theorem's Form.

By Theorem 3 and Lemma 1, $\{k_{ij}(\mathbf{x}_1, \mathbf{x}_2)\}_{i,j=1}^{V}$ form the covariance functions of a multivariate harmonizable process if and only if the associated spectral density matrix $\mathbf{S}$ satisfies two conditions: (A) its diagonal entries $S_{ii}(\boldsymbol{\omega}_1, \boldsymbol{\omega}_2)$ are PSD functions, and (B) its off-diagonal entries satisfy the Hermitian symmetry $S_{ij}(\boldsymbol{\omega}_1, \boldsymbol{\omega}_2) = \overline{S_{ji}(\boldsymbol{\omega}_2, \boldsymbol{\omega}_1)}$. Accordingly, we construct $\mathbf{S}$ as follows:

$$S_{ij}(\boldsymbol{\omega}_1, \boldsymbol{\omega}_2) = \frac{1}{4} \left[ P_{ij}(\boldsymbol{\omega}_1, \boldsymbol{\omega}_2) + P_{ij}(\boldsymbol{\omega}_2, \boldsymbol{\omega}_1) + P_{ij}(\boldsymbol{\omega}_1)\delta(\boldsymbol{\omega}_2 - \boldsymbol{\omega}_1) + P_{ij}(\boldsymbol{\omega}_2)\delta(\boldsymbol{\omega}_1 - \boldsymbol{\omega}_2) \right], \tag{11}$$

where $\delta$ is the Dirac delta, $P_{ij}(\boldsymbol{\omega}_1, \boldsymbol{\omega}_2) = \overline{P_{ji}(\boldsymbol{\omega}_1, \boldsymbol{\omega}_2)}$ is a joint density function, and $P_{ij}(\boldsymbol{\omega}_1), P_{ij}(\boldsymbol{\omega}_2)$ are marginal distributions. Moreover, we denote the corresponding Lebesgue-Stieltjes measure as $F_{ij}^P$. By replacing the Eq. (11) into the duality given by Eq. (10) and Lemma 1, we have

$$k_{ij}(\mathbf{x}_1, \mathbf{x}_2) = \iint_{\mathbb{R}^D \times \mathbb{R}^D} e^{i(\boldsymbol{\omega}_1^\top \mathbf{x}_1 - \boldsymbol{\omega}_2^\top \mathbf{x}_2)} S_{ij}(\boldsymbol{\omega}_1, \boldsymbol{\omega}_2) \, d\boldsymbol{\omega}_1 d\boldsymbol{\omega}_2$$

$$= \frac{1}{4} \iint P_{ij}(\boldsymbol{\omega}_1, \boldsymbol{\omega}_2) e^{i(\boldsymbol{\omega}_1^\top \mathbf{x}_1 - \boldsymbol{\omega}_2^\top \mathbf{x}_2)} \, d\boldsymbol{\omega}_1 d\boldsymbol{\omega}_2 + \frac{1}{4} \iint P_{ij}(\boldsymbol{\omega}_2, \boldsymbol{\omega}_1) e^{i(\boldsymbol{\omega}_1^\top \mathbf{x}_1 - \boldsymbol{\omega}_2^\top \mathbf{x}_2)} \, d\boldsymbol{\omega}_1 d\boldsymbol{\omega}_2$$

$$+ \frac{1}{4} \iint P_{ij}(\boldsymbol{\omega}_1)\delta(\boldsymbol{\omega}_2 - \boldsymbol{\omega}_1) e^{i(\boldsymbol{\omega}_1^\top \mathbf{x}_1 - \boldsymbol{\omega}_2^\top \mathbf{x}_2)} \, d\boldsymbol{\omega}_1 d\boldsymbol{\omega}_2$$

$$+ \frac{1}{4} \iint P_{ij}(\boldsymbol{\omega}_2)\delta(\boldsymbol{\omega}_1 - \boldsymbol{\omega}_2) e^{i(\boldsymbol{\omega}_1^\top \mathbf{x}_1 - \boldsymbol{\omega}_2^\top \mathbf{x}_2)} \, d\boldsymbol{\omega}_1 d\boldsymbol{\omega}_2$$

(Swap $\boldsymbol{\omega}_1 \leftrightarrow \boldsymbol{\omega}_2$ in 2nd term, integrate $\delta$ in 3rd and 4th terms)

$$= \frac{1}{4} \iint P_{ij}(\boldsymbol{\omega}_1, \boldsymbol{\omega}_2)e^{i(\boldsymbol{\omega}_1^\top \mathbf{x}_1 - \boldsymbol{\omega}_2^\top \mathbf{x}_2)} \, d\boldsymbol{\omega}_1 d\boldsymbol{\omega}_2 + \frac{1}{4} \iint P_{ij}(\boldsymbol{\omega}_1, \boldsymbol{\omega}_2)e^{i(\boldsymbol{\omega}_2^\top \mathbf{x}_1 - \boldsymbol{\omega}_1^\top \mathbf{x}_2)} \, d\boldsymbol{\omega}_1 d\boldsymbol{\omega}_2$$

$$+ \frac{1}{4} \int P_{ij}(\boldsymbol{\omega}_1)e^{i\boldsymbol{\omega}_1^\top (\mathbf{x}_1 - \mathbf{x}_2)} \, d\boldsymbol{\omega}_1 + \frac{1}{4} \int P_{ij}(\boldsymbol{\omega}_2)e^{i\boldsymbol{\omega}_2^\top (\mathbf{x}_1 - \mathbf{x}_2)} \, d\boldsymbol{\omega}_2$$

(Replacing the marginal density by the joint density for factorization)

$$= \frac{1}{4} \iint \left[ e^{i(\boldsymbol{\omega}_1^\top \mathbf{x}_1 - \boldsymbol{\omega}_2^\top \mathbf{x}_2)} + e^{i(\boldsymbol{\omega}_2^\top \mathbf{x}_1 - \boldsymbol{\omega}_1^\top \mathbf{x}_2)} + e^{i\boldsymbol{\omega}_1^\top (\mathbf{x}_1 - \mathbf{x}_2)} + e^{i\boldsymbol{\omega}_2^\top (\mathbf{x}_1 - \mathbf{x}_2)} \right] \times P_{ij}(\boldsymbol{\omega}_1, \boldsymbol{\omega}_2) \, d\boldsymbol{\omega}_1 d\boldsymbol{\omega}_2.$$

Finally, by setting the $F_{ij}^P$ as Lebesgue-Stieltjes measures associated with the spectral density functions $P_{ij}(\boldsymbol{\omega}_1, \boldsymbol{\omega}_2)$, we can express the kernel as :

$$k_{ij}(\mathbf{x}_1, \mathbf{x}_2) = \frac{1}{4} \iint [\exp(i(\boldsymbol{\omega}_1 \mathbf{x}_1 - \boldsymbol{\omega}_2 \mathbf{x}_2)) + \exp(i(\boldsymbol{\omega}_2 \mathbf{x}_1 - \boldsymbol{\omega}_1 \mathbf{x}_2)) \qquad (12)$$

$$+ \exp(i\boldsymbol{\omega}_2(\mathbf{x}_1 - \mathbf{x}_2)) + \exp(i\boldsymbol{\omega}_1(\mathbf{x}_1 - \mathbf{x}_2))]dF_{ij}^P(\boldsymbol{\omega}_1, \boldsymbol{\omega}_2) \qquad (13)$$

where $P_{ij}(\boldsymbol{\omega}_1, \boldsymbol{\omega}_2) = \overline{P_{ji}(\boldsymbol{\omega}_1, \boldsymbol{\omega}_2)}$. This result is aligned with Theorem 2.

**Sufficiency ($\Leftarrow$):** From the Theorem's Form to a Valid Kernel.

Suppose we construct the $(i, j)$-th element of kernel $k_{ij}(\mathbf{x}_1, \mathbf{x}_2)$ using the formula:

$$k_{ij}(\mathbf{x}_1, \mathbf{x}_2) = \frac{1}{4} \iint \left[ e^{i(\boldsymbol{\omega}_1^\top \mathbf{x}_1 - \boldsymbol{\omega}_2^\top \mathbf{x}_2)} + e^{i(\boldsymbol{\omega}_2^\top \mathbf{x}_1 - \boldsymbol{\omega}_1^\top \mathbf{x}_2)} + e^{i\boldsymbol{\omega}_1^\top (\mathbf{x}_1 - \mathbf{x}_2)} + e^{i\boldsymbol{\omega}_2^\top (\mathbf{x}_1 - \mathbf{x}_2)} \right] dF_{ij}^P(\boldsymbol{\omega}_1, \boldsymbol{\omega}_2)$$

where the measure $F_{ij}^P$ is associated with a density $P_{ij}$ such that $dF_{ij}^P(\boldsymbol{\omega}_1, \boldsymbol{\omega}_2) = P_{ij}(\boldsymbol{\omega}_1, \boldsymbol{\omega}_2) \, d\boldsymbol{\omega}_1 d\boldsymbol{\omega}_2$, where $P_{ij}(\boldsymbol{\omega}_1, \boldsymbol{\omega}_2) = \overline{P_{ji}(\boldsymbol{\omega}_1, \boldsymbol{\omega}_2)}$.

To establish that $k_{ij}$ defines a valid kernel for a harmonizable multivariate stochastic process, it suffices to verify that the constructed kernel satisfies conditions (A) and (B) in Lemma 1.

**Step 1: Proof of Conditions (A).**

By reversing the factorization from the necessity proof, we can write $k_{ij}$ as the Fourier transform of a spectral density $S_{ij}$.

$$k_{ij}(\mathbf{x}_1, \mathbf{x}_2) = \frac{1}{4} \iint P_{ij}(\boldsymbol{\omega}_1, \boldsymbol{\omega}_2)e^{i(\boldsymbol{\omega}_1^\top \mathbf{x}_1 - \boldsymbol{\omega}_2^\top \mathbf{x}_2)} \, d\boldsymbol{\omega}_1 d\boldsymbol{\omega}_2$$

$$+ \frac{1}{4} \iint P_{ij}(\boldsymbol{\omega}_2, \boldsymbol{\omega}_1)e^{i(\boldsymbol{\omega}_1^\top \mathbf{x}_1 - \boldsymbol{\omega}_2^\top \mathbf{x}_2)} \, d\boldsymbol{\omega}_1 d\boldsymbol{\omega}_2$$

$$+ \frac{1}{4} \iint P_{ij}(\boldsymbol{\omega}_1)\delta(\boldsymbol{\omega}_2 - \boldsymbol{\omega}_1)e^{i(\boldsymbol{\omega}_1^\top \mathbf{x}_1 - \boldsymbol{\omega}_2^\top \mathbf{x}_2)} \, d\boldsymbol{\omega}_1 d\boldsymbol{\omega}_2$$

$$+ \frac{1}{4} \iint P_{ij}(\boldsymbol{\omega}_2)\delta(\boldsymbol{\omega}_1 - \boldsymbol{\omega}_2)e^{i(\boldsymbol{\omega}_1^\top \mathbf{x}_1 - \boldsymbol{\omega}_2^\top \mathbf{x}_2)} \, d\boldsymbol{\omega}_1 d\boldsymbol{\omega}_2.$$

We can then express $k_{ij}(\mathbf{x}_1, \mathbf{x}_2) = \iint e^{i(\boldsymbol{\omega}_1^\top \mathbf{x}_1 - \boldsymbol{\omega}_2^\top \mathbf{x}_2)} S_{ij}(\boldsymbol{\omega}_1, \boldsymbol{\omega}_2) \, d\boldsymbol{\omega}_1 d\boldsymbol{\omega}_2$, where the spectral density is:

$$S_{ij}(\boldsymbol{\omega}_1, \boldsymbol{\omega}_2) = \frac{1}{4} \left[ P_{ij}(\boldsymbol{\omega}_1, \boldsymbol{\omega}_2) + P_{ij}(\boldsymbol{\omega}_2, \boldsymbol{\omega}_1) + P_{ij}(\boldsymbol{\omega}_1)\delta(\boldsymbol{\omega}_2 - \boldsymbol{\omega}_1) + P_{ij}(\boldsymbol{\omega}_2)\delta(\boldsymbol{\omega}_1 - \boldsymbol{\omega}_2) \right],$$

which is a PSD function for $S_{ii}$, and thus condition (A) is satisfied.

**Step 2: Proof of Conditions (B).**

It remains to verify the Hermitian symmetry $S_{ij}(\boldsymbol{\omega}_1, \boldsymbol{\omega}_2) = \overline{S_{ji}(\boldsymbol{\omega}_2, \boldsymbol{\omega}_1)}$. We start from the right-hand side:

$$\overline{S_{ji}(\boldsymbol{\omega}_2, \boldsymbol{\omega}_1)} = \frac{1}{4}\left[\overline{P_{ji}(\boldsymbol{\omega}_2, \boldsymbol{\omega}_1)} + \overline{P_{ji}(\boldsymbol{\omega}_1, \boldsymbol{\omega}_2)} + \overline{P_{ji}(\boldsymbol{\omega}_2)}\delta(\boldsymbol{\omega}_1 - \boldsymbol{\omega}_2) + \overline{P_{ji}(\boldsymbol{\omega}_1)}\delta(\boldsymbol{\omega}_2 - \boldsymbol{\omega}_1)\right].$$

Using our core assumption that $P_{ij}(\boldsymbol{\omega}_1, \boldsymbol{\omega}_2) = \overline{P_{ji}(\boldsymbol{\omega}_1, \boldsymbol{\omega}_2)}$, this implies for the marginals that $P_{ij}(\boldsymbol{\omega}) = \overline{P_{ji}(\boldsymbol{\omega})}$. Substituting these into the expression for $S_{ji}(\boldsymbol{\omega}_2, \boldsymbol{\omega}_1)$:

$$\begin{aligned}
S_{ij}(\boldsymbol{\omega}_2, \boldsymbol{\omega}_1) &= \frac{1}{4}\left[P_{ij}(\boldsymbol{\omega}_1, \boldsymbol{\omega}_2) + P_{ij}(\boldsymbol{\omega}_2, \boldsymbol{\omega}_1) + P_{ij}(\boldsymbol{\omega}_2)\delta(\boldsymbol{\omega}_1 - \boldsymbol{\omega}_2) + P_{ij}(\boldsymbol{\omega}_1)\delta(\boldsymbol{\omega}_2 - \boldsymbol{\omega}_1)\right] \\
&= \overline{S_{ji}(\boldsymbol{\omega}_2, \boldsymbol{\omega}_1)}.
\end{aligned}$$

The Hermitian condition (B) holds.

Since the constructed kernel $k_{ij}$ possesses a spectral density $S_{ij}$ that satisfies the conditions given by Lemma 1, $k_{ij}$ is a valid covariance kernel for a a harmonizable multivariate stochastic process. This completes the proof.

## A.2 Derivation of the Multi-output Low-Rank Nonstationary Kernel

Before detailing the kernel construction, we first introduce two mathematical identities that are essential for the derivation.

**Identity 1** (Product of Gaussian). *Let $p_i(\mathbf{w})$ and $p_j(\mathbf{w})$ be two $2D$-dimensional multivariate normal distributions:*

$$p_i(\mathbf{w}) = \frac{1}{(2\pi)^D |\boldsymbol{\Sigma}|_i^{1/2}} \exp\left(-\frac{1}{2}(\mathbf{w} - \boldsymbol{\mu}_i)^\top \boldsymbol{\Sigma}_i^{-1}(\mathbf{w} - \boldsymbol{\mu}_i)\right),$$

$$p_j(\mathbf{w}) = \frac{1}{(2\pi)^D |\boldsymbol{\Sigma}|_j^{1/2}} \exp\left(-\frac{1}{2}(\mathbf{w} - \boldsymbol{\mu}_j)^\top \boldsymbol{\Sigma}_j^{-1}(\mathbf{w} - \boldsymbol{\mu}_j)\right).$$

Then, the product of these two densities is proportional to another multivariate normal distribution:

$$p_i(\mathbf{w})p_j(\mathbf{w}) = z_{ij}\,\mathcal{N}(\mathbf{w} \mid \boldsymbol{m}_{ij}, \boldsymbol{S}_{ij}),$$

where the resulting mean, covariance, and normalizing constant are

$$\boldsymbol{S}_{ij} = (\boldsymbol{\Sigma}_i^{-1} + \boldsymbol{\Sigma}_j^{-1})^{-1}, \quad \boldsymbol{m}_{ij} = \boldsymbol{S}_{ij}(\boldsymbol{\Sigma}_i^{-1}\boldsymbol{\mu}_i + \boldsymbol{\Sigma}_j^{-1}\boldsymbol{\mu}_j),$$

$$z_{ij} = (2\pi)^{-D}|\boldsymbol{\Sigma}_i + \boldsymbol{\Sigma}_j|^{-\frac{1}{2}} \exp\left\{-\frac{1}{2}(\boldsymbol{\mu}_i - \boldsymbol{\mu}_j)^\top(\boldsymbol{\Sigma}_i + \boldsymbol{\Sigma}_j)^{-1}(\boldsymbol{\mu}_i - \boldsymbol{\mu}_j)\right\}.$$

**Identity 2** (Characteristic Function). *For a random vector $\mathbf{z} \sim \mathcal{N}(\boldsymbol{\mu}, \boldsymbol{\Sigma})$, its characteristic function is*

$$\phi(\mathbf{t}) = E[e^{i\mathbf{z}^\top \mathbf{t}}] = \int e^{i\mathbf{z}^\top \mathbf{t}} \mathcal{N}(\mathbf{z}|\boldsymbol{\mu}, \boldsymbol{\Sigma})\,d\mathbf{z} = \exp\left(i\boldsymbol{\mu}^\top \mathbf{t} - \frac{1}{2}\mathbf{t}^\top \boldsymbol{\Sigma}\mathbf{t}\right).$$

This formula enables us to bypass direct integration and obtain a closed-form analytical result.

**Step 1: Spectral Density Construction.**

As discussed in § 3.2, we propose a spectral density design that is both parameter-efficient and expressive. For each output $i$, we assign a latent vector $\mathbf{r}_i \in \mathbb{R}^Q$ and define $P_{ij}(\boldsymbol{\omega}_1, \boldsymbol{\omega}_2) = \mathbf{r}_i^H \mathbf{r}_j$. And each component $r_i^{(q)}$ of $\mathbf{r}_i$ is modeled as a bivariate Gaussian distribution:

$$r_i^{(q)}(\boldsymbol{\omega}_1, \boldsymbol{\omega}_2) = w_i^{(q)}\mathcal{N}\left(\begin{pmatrix}\boldsymbol{\omega}_1 \\ \boldsymbol{\omega}_2\end{pmatrix}\middle|\; \boldsymbol{\mu}_i^{(q)}, \boldsymbol{\Sigma}_i^{(q)}\right),$$

where $w_i^{(q)} \in \mathbb{R}^+$ is a positive weight, and the mean vector and covariance matrix are partitioned as:

$$\boldsymbol{\mu}_i^{(q)} = \begin{pmatrix}\boldsymbol{\mu}_{i1}^{(q)} \\ \boldsymbol{\mu}_{i2}^{(q)}\end{pmatrix}, \quad \boldsymbol{\Sigma}_i^{(q)} = \begin{bmatrix}\boldsymbol{\Sigma}_{i1}^{(q)} & (\boldsymbol{\Sigma}_{ic}^{(q)})^\top \\ \boldsymbol{\Sigma}_{ic}^{(q)} & \boldsymbol{\Sigma}_{i2}^{(q)}\end{bmatrix}.$$

The covariance matrices are parameterized by $\boldsymbol{\Sigma}_{i1}^{(q)} = \text{diag}((\boldsymbol{\sigma}_{i1}^{(q)})^2)$, $\boldsymbol{\Sigma}_{i2}^{(q)} = \text{diag}((\boldsymbol{\sigma}_{i2}^{(q)})^2)$, and the cross-covariance by $\boldsymbol{\Sigma}_{ic}^{(q)} = \rho_i^{(q)} \text{diag}(\boldsymbol{\sigma}_{i1}^{(q)}) \text{diag}(\boldsymbol{\sigma}_{i2}^{(q)})$, where $\boldsymbol{\sigma}_{i1}^{(q)}, \boldsymbol{\sigma}_{i2}^{(q)} \in \mathbb{R}^D$, and $\rho_i^{(q)} \in [-1, 1]$ denotes the correlation coefficient.

By Identity 1, the spectral density has the following form:

$$
\begin{aligned}
P_{ij}(\boldsymbol{\omega}_1, \boldsymbol{\omega}_2) =& \mathbf{r}_i^H \mathbf{r}_j = \sum_{q=1}^Q \overline{r_i^{(q)}} r_j^{(q)} \\
=& \sum_{q=1}^Q P_{ij}^{(q)}(\boldsymbol{\omega}_1, \boldsymbol{\omega}_2) = \sum_{q=1}^Q z_{ij}^{(q)} \mathcal{N}\left( \begin{pmatrix} \boldsymbol{\omega}_1 \\ \boldsymbol{\omega}_2 \end{pmatrix} \Big| \, \boldsymbol{m}_{ij}^{(q)}, \, \boldsymbol{S}_{ij}^{(q)} \right),
\end{aligned}
$$

where the parameters of each mixture component are derived as:

$$
\begin{aligned}
\boldsymbol{S}_{ij}^{(q)} &= \left( (\boldsymbol{\Sigma}_i^{(q)})^{-1} + (\boldsymbol{\Sigma}_j^{(q)})^{-1} \right)^{-1}, \\
\boldsymbol{m}_{ij}^{(q)} &= \boldsymbol{S}_{ij}^{(q)} \left( (\boldsymbol{\Sigma}_i^{(q)})^{-1} \boldsymbol{\mu}_i^{(q)} + (\boldsymbol{\Sigma}_j^{(q)})^{-1} \boldsymbol{\mu}_j^{(q)} \right), \\
z_{ij}^{(q)} &= w_i^{(q)} w_j^{(q)} (2\pi)^{-D} |\boldsymbol{\Sigma}_i^{(q)} + \boldsymbol{\Sigma}_j^{(q)}|^{-\frac{1}{2}} \exp\left\{ -\frac{1}{2} (\boldsymbol{\mu}_i^{(q)} - \boldsymbol{\mu}_j^{(q)})^\top (\boldsymbol{\Sigma}_i^{(q)} + \boldsymbol{\Sigma}_j^{(q)})^{-1} (\boldsymbol{\mu}_i^{(q)} - \boldsymbol{\mu}_j^{(q)}) \right\}.
\end{aligned}
$$

Since we only focus on real-valued kernels, we eliminate the imaginary part by enforcing a symmetric spectral density:

$$
\begin{aligned}
P_{ij}^{\text{symm}}(\boldsymbol{\omega}_1, \boldsymbol{\omega}_2) &= \sum_{q=1}^Q P_{ij}^{\text{symm},(q)}(\boldsymbol{\omega}_1, \boldsymbol{\omega}_2) \\
&= \sum_{q=1}^Q \frac{1}{2} \left[ P_{ij}^{(q)}(\boldsymbol{\omega}_1, \boldsymbol{\omega}_2) + P_{ij}^{(q)}(-\boldsymbol{\omega}_1, -\boldsymbol{\omega}_2) \right] \\
&= \sum_{q=1}^Q \frac{z_{ij}^q}{2} \left[ \mathcal{N}\left( \begin{pmatrix} \boldsymbol{\omega}_1 \\ \boldsymbol{\omega}_2 \end{pmatrix} \Big| \boldsymbol{m}_{ij}^{(q)}, \boldsymbol{S}_{ij}^{(q)} \right) + \mathcal{N}\left( \begin{pmatrix} \boldsymbol{\omega}_1 \\ \boldsymbol{\omega}_2 \end{pmatrix} \Big| - \boldsymbol{m}_{ij}^{(q)}, \boldsymbol{S}_{ij}^{(q)} \right) \right].
\end{aligned}
$$

**Step 2: Transform Spectral Density to the Kernel Domain.**

We now transform spectral density into the kernel domain by Theorem 2, yielding the $(i, j)$-th element of kernels $k_{ij}(\mathbf{x}_1, \mathbf{x}_2)$

$$
\begin{aligned}
&= \frac{1}{4} \iint \left[ e^{i(\boldsymbol{\omega}_1^\top \mathbf{x}_1 - \boldsymbol{\omega}_2^\top \mathbf{x}_2)} + e^{i(\boldsymbol{\omega}_2^\top \mathbf{x}_1 - \boldsymbol{\omega}_1^\top \mathbf{x}_2)} + e^{i\boldsymbol{\omega}_1^\top (\mathbf{x}_1 - \mathbf{x}_2)} + e^{i\boldsymbol{\omega}_2^\top (\mathbf{x}_1 - \mathbf{x}_2)} \right] P_{ij}^{\text{symm}} \, d\boldsymbol{\omega}_1 d\boldsymbol{\omega}_2 \\
&= \sum_{q=1}^Q \left[ \frac{1}{4} \iint \left[ e^{i(\boldsymbol{\omega}_1^\top \mathbf{x}_1 - \boldsymbol{\omega}_2^\top \mathbf{x}_2)} + e^{i(\boldsymbol{\omega}_2^\top \mathbf{x}_1 - \boldsymbol{\omega}_1^\top \mathbf{x}_2)} + e^{i\boldsymbol{\omega}_1^\top (\mathbf{x}_1 - \mathbf{x}_2)} + e^{i\boldsymbol{\omega}_2^\top (\mathbf{x}_1 - \mathbf{x}_2)} \right] P_{ij}^{\text{symm},(q)} \, d\boldsymbol{\omega}_1 d\boldsymbol{\omega}_2 \right] \\
&= \sum_{q=1}^Q \Bigg[ \underbrace{\frac{1}{4} \iint e^{i(\boldsymbol{\omega}_1^\top \mathbf{x}_1 - \boldsymbol{\omega}_2^\top \mathbf{x}_2)} P_{ij}^{\text{symm},(q)} \, d\boldsymbol{\omega}_1 d\boldsymbol{\omega}_2}_{\text{Term 1} \, (I_1^{(q)})} + \underbrace{\frac{1}{4} \iint e^{i(-\boldsymbol{\omega}_1^\top \mathbf{x}_2 + \boldsymbol{\omega}_2^\top \mathbf{x}_1)} P_{ij}^{\text{symm},(q)} \, d\boldsymbol{\omega}_1 d\boldsymbol{\omega}_2}_{\text{Term 2} \, (I_2^{(q)})} \\
&\quad + \underbrace{\frac{1}{4} \iint e^{i\boldsymbol{\omega}_1^\top (\mathbf{x}_1 - \mathbf{x}_2)} P_{ij}^{\text{symm},(q)} \, d\boldsymbol{\omega}_1 d\boldsymbol{\omega}_2}_{\text{Term 3} \, (I_3^{(q)})} + \underbrace{\frac{1}{4} \iint e^{i\boldsymbol{\omega}_2^\top (\mathbf{x}_1 - \mathbf{x}_2)} P_{ij}^{\text{symm},(q)} \, d\boldsymbol{\omega}_1 d\boldsymbol{\omega}_2}_{\text{Term 4} \, (I_4^{(q)})} \Bigg].
\end{aligned}
$$

In fact, terms 1–4 correspond to the characteristic function of a Gaussian distribution. Specifically, we can derive their general analytic form by setting $\boldsymbol{\omega} = \begin{pmatrix} \boldsymbol{\omega}_1 \\ \boldsymbol{\omega}_2 \end{pmatrix}$ and $\mathbf{t} = \begin{pmatrix} \mathbf{t}_1 \\ \mathbf{t}_2 \end{pmatrix}$.

$$
\int e^{i\boldsymbol{\omega}^\top \mathbf{t}} P_{ij}^{\text{symm},(q)}(\boldsymbol{\omega}) \, d\boldsymbol{\omega} = \int e^{i\boldsymbol{\omega}^\top \mathbf{t}} \frac{z_{ij}^{(q)}}{2} \left[ \mathcal{N}(\boldsymbol{\omega} | \boldsymbol{m}_{ij}^{(q)}, \boldsymbol{S}_{ij}^{(q)}) + \mathcal{N}(\boldsymbol{\omega} | -\boldsymbol{m}_{ij}^{(q)}, \boldsymbol{S}_{ij}^{(q)}) \right] d\boldsymbol{\omega}
$$

$$
= \frac{z_{ij}^{(q)}}{2} \left[ \int e^{i\boldsymbol{\omega}^\top \mathbf{t}} \mathcal{N}(\boldsymbol{\omega} | \boldsymbol{m}_{ij}^{(q)}, \boldsymbol{S}_{ij}^{(q)}) d\boldsymbol{\omega} + \int e^{i\boldsymbol{\omega}^\top \mathbf{t}} \mathcal{N}(\boldsymbol{\omega} | -\boldsymbol{m}_{ij}^{(q)}, \boldsymbol{S}_{ij}^{(q)}) d\boldsymbol{\omega} \right]
$$

$$
= \frac{z_{ij}^{(q)}}{2} \left[ \exp\left( i(\boldsymbol{m}_{ij}^{(q)})^\top \mathbf{t} - \frac{1}{2}\mathbf{t}^\top \boldsymbol{S}_{ij}^{(q)} \mathbf{t} \right) + \exp\left( -i(\boldsymbol{m}_{ij}^{(q)})^\top \mathbf{t} - \frac{1}{2}\mathbf{t}^\top \boldsymbol{S}_{ij}^{(q)} \mathbf{t} \right) \right] \text{ (By Identity 2)}
$$

$$
= z_{ij}^{(q)} \cos\left( (\boldsymbol{m}_{ij}^{(q)})^\top \mathbf{t} \right) \exp\left( -\frac{1}{2}\mathbf{t}^\top \boldsymbol{S}_{ij}^{(q)} \mathbf{t} \right).
$$

Thus, we can use this result to derive the analytic form of terms 1–4 as following.

**Term 1:** By setting $\mathbf{t} = \begin{pmatrix} \mathbf{x}_1 \\ -\mathbf{x}_2 \end{pmatrix}$, The result is:

$$
I_1^{(q)} = \frac{z_{ij}^{(q)}}{4} \cos\left( (\boldsymbol{m}_{ij1}^{(q)})^\top \mathbf{x}_1 - (\boldsymbol{m}_{ij2}^{(q)})^\top \mathbf{x}_2 \right) \exp\left( -\frac{1}{2}(\mathbf{x}_1^\top \boldsymbol{S}_{ij1}^{(q)} \mathbf{x}_1 - 2\mathbf{x}_1^\top (\boldsymbol{S}_{ijc}^{(q)})^\top \mathbf{x}_2 + \mathbf{x}_2^\top \boldsymbol{S}_{ij2}^{(q)} \mathbf{x}_2) \right).
$$

**Term 2:** By setting $\mathbf{t} = \begin{pmatrix} -\mathbf{x}_2 \\ \mathbf{x}_1 \end{pmatrix}$. The result is:

$$
I_2^{(q)} = \frac{z_{ij}^{(q)}}{4} \cos\left( (\boldsymbol{m}_{ij2}^{(q)})^\top \mathbf{x}_1 - (\boldsymbol{m}_{ij1}^{(q)})^\top \mathbf{x}_2 \right) \exp\left( -\frac{1}{2}(\mathbf{x}_2^\top \boldsymbol{S}_{ij1}^{(q)} \mathbf{x}_2 - 2\mathbf{x}_1^\top \boldsymbol{S}_{ijc}^{(q)} \mathbf{x}_2 + \mathbf{x}_1^\top \boldsymbol{S}_{ij2}^{(q)} \mathbf{x}_1) \right).
$$

**Term 3:** By setting $\mathbf{t} = \begin{pmatrix} \mathbf{x}_1 - \mathbf{x}_2 \\ \mathbf{0} \end{pmatrix}$. The result is:

$$
I_3^{(q)} = \frac{z_{ij}^{(q)}}{4} \cos\left( (\boldsymbol{m}_{ij1}^{(q)})^\top (\mathbf{x}_1 - \mathbf{x}_2) \right) \exp\left( -\frac{1}{2}(\mathbf{x}_1 - \mathbf{x}_2)^\top \boldsymbol{S}_{ij1}^{(q)} (\mathbf{x}_1 - \mathbf{x}_2) \right).
$$

**Term 4:** By setting $\mathbf{t} = \begin{pmatrix} \mathbf{0} \\ \mathbf{x}_1 - \mathbf{x}_2 \end{pmatrix}$. The result is:

$$
I_4^{(q)} = \frac{z_{ij}^{(q)}}{4} \cos\left( (\boldsymbol{m}_{ij2}^{(q)})^\top (\mathbf{x}_1 - \mathbf{x}_2) \right) \exp\left( -\frac{1}{2}(\mathbf{x}_1 - \mathbf{x}_2)^\top \boldsymbol{S}_{ij2}^{(q)} (\mathbf{x}_1 - \mathbf{x}_2) \right).
$$

where $\boldsymbol{m}_{ij}^{(q)} = \begin{pmatrix} \boldsymbol{m}_{ij1}^{(q)} \\ \boldsymbol{m}_{ij2}^{(q)} \end{pmatrix}$ and $\boldsymbol{S}_{ij}^{(q)} = \begin{bmatrix} \boldsymbol{S}_{ij1}^{(q)} & (\boldsymbol{S}_{ijc}^{(q)})^\top \\ \boldsymbol{S}_{ijc}^{(q)} & \boldsymbol{S}_{ij2}^{(q)} \end{bmatrix}$.

Thus, the explicit form of the $q$-th component of $k_{ij}(\mathbf{x}_1, \mathbf{x}_2)$ can be derived by combining four terms. Finally, summing over all $Q$ components yields the MO-LRN kernel as below.

**Definition 2** (Multi-output Low-Rank Nonstationary Kernel). *The MO-LRN kernel for Multi-output Gaussian Process with component parameters $(z_{ij}^{(q)}, \boldsymbol{m}_{ij}^{(q)}, \boldsymbol{S}_{ij}^{(q)})$ for $q \in \{1, \dots, Q\}$. The parameters are partitioned as:*

$$
\boldsymbol{m}_{ij}^{(q)} = \begin{pmatrix} \boldsymbol{m}_{ij1}^{(q)} \\ \boldsymbol{m}_{ij2}^{(q)} \end{pmatrix} \quad \text{and} \quad \boldsymbol{S}_{ij}^{(q)} = \begin{bmatrix} \boldsymbol{S}_{ij1}^{(q)} & (\boldsymbol{S}_{ijc}^{(q)})^\top \\ \boldsymbol{S}_{ijc}^{(q)} & \boldsymbol{S}_{ij2}^{(q)} \end{bmatrix}.
$$

*The resulting cross-covariance kernel $k_{ij}(\mathbf{x}_1, \mathbf{x}_2)$ is given by the explicit formula:*

$$
k_{ij}(\mathbf{x}_1, \mathbf{x}_2) = \sum_{q=1}^{Q} \frac{z_{ij}^{(q)}}{4} \left[ \cos\left( \alpha_1^{(q)} \right) \exp\left( -\frac{1}{2}\beta_1^{(q)} \right) + \cos\left( \alpha_2^{(q)} \right) \exp\left( -\frac{1}{2}\beta_2^{(q)} \right) \right.
$$

$$
\left. + \cos\left( \alpha_3^{(q)} \right) \exp\left( -\frac{1}{2}\beta_3^{(q)} \right) + \cos\left( \alpha_4^{(q)} \right) \exp\left( -\frac{1}{2}\beta_4^{(q)} \right) \right].
$$

where the auxiliary functions $\alpha_k^{(q)}$ and $\beta_k^{(q)}$ are defined for each component $q$ as:

$$\alpha_1^{(q)} = (\boldsymbol{m}_{ij1}^{(q)})^\top \mathbf{x}_1 - (\boldsymbol{m}_{ij2}^{(q)})^\top \mathbf{x}_2,$$

$$\beta_1^{(q)} = \mathbf{x}_1^\top \boldsymbol{S}_{ij1}^{(q)} \mathbf{x}_1 - 2\mathbf{x}_1^\top (\boldsymbol{S}_{ijc}^{(q)})^\top \mathbf{x}_2 + \mathbf{x}_2^\top \boldsymbol{S}_{ij2}^{(q)} \mathbf{x}_2,$$

$$\alpha_2^{(q)} = (\boldsymbol{m}_{ij2}^{(q)})^\top \mathbf{x}_1 - (\boldsymbol{m}_{ij1}^{(q)})^\top \mathbf{x}_2,$$

$$\beta_2^{(q)} = \mathbf{x}_2^\top \boldsymbol{S}_{ij1}^{(q)} \mathbf{x}_2 - 2\mathbf{x}_1^\top \boldsymbol{S}_{ijc}^{(q)} \mathbf{x}_2 + \mathbf{x}_1^\top \boldsymbol{S}_{ij2}^{(q)} \mathbf{x}_1,$$

$$\alpha_3^{(q)} = (\boldsymbol{m}_{ij1}^{(q)})^\top (\mathbf{x}_1 - \mathbf{x}_2),$$

$$\beta_3^{(q)} = (\mathbf{x}_1 - \mathbf{x}_2)^\top \boldsymbol{S}_{ij1}^{(q)} (\mathbf{x}_1 - \mathbf{x}_2),$$

$$\alpha_4^{(q)} = (\boldsymbol{m}_{ij2}^{(q)})^\top (\mathbf{x}_1 - \mathbf{x}_2),$$

$$\beta_4^{(q)} = (\mathbf{x}_1 - \mathbf{x}_2)^\top \boldsymbol{S}_{ij2}^{(q)} (\mathbf{x}_1 - \mathbf{x}_2).$$

The parameters of the kernel are ultimately determined by the set of hyperparameters $\boldsymbol{\theta} = \left\{ w_i^{(q)}, \boldsymbol{\mu}_{i1}^{(q)}, \boldsymbol{\mu}_{i2}^{(q)}, \boldsymbol{\sigma}_{i1}^{(q)}, \boldsymbol{\sigma}_{i2}^{(q)}, \rho_i^{(q)} \right\}_{q=1,i=1}^{Q,V}$.

# B EXPERIMENTAL DETAILS

## B.1 DATASET DESCRIPTION

This subsection details the datasets employed in our experiments and the comprehensive preprocessing method.

### B.1.1 SYNTHETIC DATASET

The synthetic data is generated from two functions that exhibit input-dependent variation:

**1) Piecewise Function:**

$$f^{(1)}(x) = \begin{cases} \sin(2\pi x) + \epsilon & \text{if } x < -1 \\ \sin(4\pi x) + \epsilon & \text{if } -1 \le x < 1 \\ 0.5x(\sin(6\pi x) + \sin(2\pi x)) + \epsilon & \text{if } x \ge 1 \end{cases} \tag{14}$$

**2) Periodic Function:**

$$f^{(2)}(x) = A(x)\sin(2\pi \cdot f(x) \cdot x) + \epsilon, \tag{15}$$

where $A(x) = 0.5 + 1.5 \cdot \frac{x}{6}$ is the amplitude function that increases with $x$, $f(x) = 3 - 2 \cdot \frac{x}{6}$ is the frequency function that decreases with $x$, and $\epsilon \sim \mathcal{N}(0, 0.1)$ represents Gaussian noise.

The final dataset comprises 400 input–output pairs over the domain $x \in [0, 5]$, where each input $x_n$ is associated with the two-dimensional output $\mathbf{y}_n = [f^{(1)}(x_n), f^{(2)}(x_n)]^\top$. We randomly split the dataset into 70% for training and 30% for testing, and standardize both output dimensions before model fitting.

### B.1.2 ELECTRICITY TRANSFORMER TEMPERATURE (ETT)

The electricity transformer temperature (ETT) dataset[4] (Zhou et al., 2021) records transformer data over two years (July 2016–June 2018) at 15-minute intervals, yielding 69,680 samples with seven variables: oil temperature and six power load measurements. For our experiments, we selected a one-week subset (July 1–7, 2016), giving 670 samples across all seven variables. We randomly split the data into training (70%) and test (30%) sets with shuffling. All variables were standardized to zero mean and unit variance. The timestamps were converted into minutes and rescaled to $[0, 600]$ using min–max normalization, preserving relative time intervals while reducing scale. Each standardized variable was treated as an output channel, enabling multi-output modeling and joint prediction of all seven transformer metrics.

---

[4]https://github.com/zhouhaoyi/ETDataset

**Table 5:** Hyperparameter settings for the MO-LRN kernel.

| Kernel Setup | | | |
|---|---|---|---|
| **Parameters** | **Synthetic Data** | **ETT** | **Air Quality** |
| # Mixture densities ($Q$) | 2 | 4 | 4 |
| **Optimizer Setup (Adam)** | | | |
| **Parameters** | **Synthetic Data** | **ETT** | **Air Quality** |
| Learning rate | 0.10 | 0.02 | 0.01 |
| Weight decay | $10^{-5}$ | N/A | N/A |
| # Iterations | 500 | 4000 | 4000 |

**Table 6:** Consolidated hyperparameter settings for the ETT and Airquality datasets across various models.

| ETT Dataset | | | | | |
|---|---|---|---|---|---|
| **Model** | **I** | **P** | **Q** | **lr** | **# Iterations** |
| LMC-NGSM | 4 | N/A | 4 | 0.02 | 4000 |
| LMC-SM | 4 | N/A | 4 | 0.02 | 4000 |
| CONV | N/A | N/A | 4 | 0.02 | 4000 |
| MOSM | N/A | N/A | 4 | 0.02 | 4000 |
| MOHSM | N/A | 4 | 4 | 0.02 | 4000 |
| **Airquality Dataset** | | | | | |
| **Model** | **I** | **P** | **Q** | **lr** | **# Iterations** |
| LMC-NGSM | 4 | N/A | 4 | 0.01 | 4000 |
| LMC-SM | 4 | N/A | 4 | 0.01 | 4000 |
| CONV | N/A | N/A | 4 | 0.01 | 4000 |
| MOSM | N/A | N/A | 4 | 0.01 | 4000 |
| MOHSM | N/A | 4 | 4 | 0.01 | 4000 |

### B.1.3 AIR QUALITY

The air quality dataset[5] (Zhang et al., 2017) contains hourly measurements from Beijing monitoring stations between 2013 and 2017. We used one week of data (March 1–8, 2013) from the Aotizhongxin station, yielding 168 hourly samples with eight features: PM2.5, PM10, SO2, NO2, CO, O3, temperature, and pressure. Timestamps were converted to hours since March 1, 2013, and rescaled to $[0, 160]$ using min–max normalization. All features were standardized to zero mean and unit variance. For *interpolation*, we randomly removed 20% of the data points across all variables. For *imputation*, we removed contiguous intervals in different channels: PM2.5 (0–20h), PM10 (25–45h), SO2 (50–70h), NO2 (75–95h), CO (100–120h), and O3 (125–145h).

## B.2 BENCHMARK METHODS

We present a detailed discussion of the theoretical foundations of the benchmark methods summarized in Table 2. For implementation, we rely on the MOGP Toolkit (MOGPTK)[6] (de Wolff et al., 2021).

---

[5]https://archive.ics.uci.edu/dataset/501/beijing+multi+site+air+quality+data
[6]https://github.com/GAMES-UChile/mogptk

### B.2.1 LINEAR MODEL OF COREGIONALIZATION

The linear model of coregionalization (LMC) (Goovaerts, 1997) represents each output $f_v(\mathbf{x})$ as a linear combination of latent Gaussian processes:

$$f_v(\mathbf{x}) = \sum_{i=1}^{I} \sum_{j=1}^{J_i} a_{v,i}^j u_i^j(\mathbf{x}). \tag{16}$$

Here:

- $u_i^j(\mathbf{x})$ are independent latent Gaussian processes within the same group $i$ share the same covariance function $k_i(\cdot, \cdot)$.
- $a_{v,i}^j$ are the scalar coefficients that mix the latent functions to create the observed output $f_v(\mathbf{x})$.
- $I$ is the number of latent process groups (each associated with a different covariance function), and $J_i$ is the number of latent processes within each group $i$.

From the perspective of MOGP, the kernel function of LMC is given by:

$$\mathbf{K}(\mathbf{x}_1, \mathbf{x}_2) = \mathrm{cov}[\mathbf{f}(\mathbf{x}_1), \mathbf{f}(\mathbf{x}_2)] = \sum_{i=1}^{I} \mathbf{B}_i k_i(\mathbf{x}_1, \mathbf{x}_2). \tag{17}$$

where $\mathbf{B}_i$ is a $V \times V$ positive semi-definite matrix determined by the coefficients $a_{v,i}^j$. It captures the correlations between the outputs that are explained by the $i$-th group of latent functions. The rank of $\mathbf{B}_i$ is $J_i$, which controls the complexity of the correlation structure for that component.

### B.2.2 CONVOLUTION PROCESS

The process convolution method (Boyle and Frean, 2004) offers an alternative way to generate correlated outputs. Each output $f_v(\mathbf{x})$ is obtained by convolving a shared latent GP $u(\mathbf{x})$ with a smoothing kernel $G_v(\mathbf{x})$. Intuitively, this corresponds to taking a single underlying random process and "smoothing" it differently to produce the observed outputs.

Formally, the model for the $v$-th output is expressed as a convolution integral:

$$f_v(\mathbf{x}) = \int_{\mathcal{X}} G_v(\mathbf{x} - \mathbf{z}) u(\mathbf{z}) d\mathbf{z}, \tag{18}$$

where

- $u(\mathbf{x})$ is a shared latent GP with a kernel function $k(\cdot, \cdot)$.
- $G_v(\mathbf{x})$ is the smoothing kernel specific to the $v$-th output.

Based on this convolutional structure, the cross-covariance between $f_{v_i}(\mathbf{x}_p)$ and $f_{v_j}(\mathbf{x}_q)$, (i.e., $k_{ij}(\mathbf{x}_p, \mathbf{x}_q)$) is given by

$$k_{ij}(\mathbf{x}_p, \mathbf{x}_q) = \mathrm{cov}[f_{v_i}(\mathbf{x}_p), f_{v_j}(\mathbf{x}_q)] = \int_{\mathcal{X}} \int_{\mathcal{X}} G_{v_i}(\mathbf{x}_p - \mathbf{z}) G_{v_j}(\mathbf{x}_q - \mathbf{z}') k(\mathbf{z}, \mathbf{z}') d\mathbf{z} d\mathbf{z}'. \tag{19}$$

### B.2.3 MULTI-OUTPUT SPECTRAL MIXUTRE (MOSM) KERNEL

The multi-output spectral mixture (MOSM) (Parra and Tobar, 2017) kernel is constructed by designing the spectral density and then mapping it back to the kernel space via the well-established duality for stationary MOGP kernel. Its form is given by:

$$k_{ij}(\boldsymbol{\tau}) = \sum_{q=1}^{Q} \alpha_{ij}^{(q)} \exp\left(-\frac{1}{2}(\boldsymbol{\tau} + \boldsymbol{\theta}_{ij}^{(q)})^\top \boldsymbol{\Sigma}_{ij}^{(q)} (\boldsymbol{\tau} + \boldsymbol{\theta}_{ij}^{(q)})\right) \cos\left((\boldsymbol{\tau} + \boldsymbol{\theta}_{ij}^{(q)})^\top \boldsymbol{\mu}_{ij}^{(q)} + \phi_{ij}^{(q)}\right). \tag{20}$$

where $\alpha_{ij}^{(q)} = w_{ij}^{(q)}(2\pi)^{n/2}|\boldsymbol{\Sigma}_{ij}^{(q)}|^{1/2}$ and the superindex $(\cdot)^{(q)}$ denotes the parameter of the $q^{th}$ component of the spectral mixture.

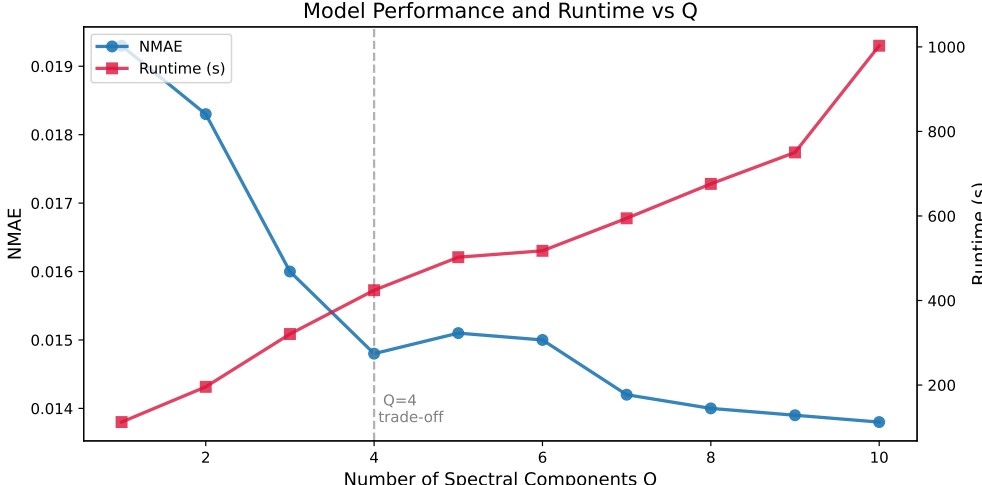

**Figure 5:** Model performance and computational cost with varying numbers of spectral components $Q$ in MO-LRN. The left $y$-axis shows the NMAE, and the right $y$-axis shows the corresponding runtime.

### B.3 HYPERPARAMETER SETTINGS

To select a reasonable value for $Q$, we evaluate both runtime and NMAE on the GOUN dataset as $Q$ varies (see Appendix C.1 for details), and present the results in Figure 5. As $Q$ increases, the model achieves slightly better predictive accuracy (lower NMAE), but at the cost of substantially longer computation time. Particularly, the accuracy gain beyond $Q = 4$ is marginal, whereas the runtime grows rapidly, suggesting that $Q = 4$ provides a favorable balance between accuracy and efficiency.

In addition, to ensure a fair comparison, we set the number of mixture components to the same value across all competitors. Moreover, in the toy example, MOHSM is also optimized for 500 iterations for consistency. The configurations for our model and the baselines are summarized in Table 5 and Table 6, respectively.

## C ADDITIONAL EXPERIMENTS

In this section, we evaluate the proposed MO-LRN kernel on MOGP regression tasks using additional datasets, all obtained from the MOGPTK package (de Wolff et al., 2021). We further include a runtime analysis on the ETT dataset with varying sample sizes to demonstrate the efficiency of our model.

### C.1 DATASET DESCRIPTION

We first provide a brief description of the datasets and then present the corresponding results.

**GONU dataset:** A real-world financial dataset consisting of weekly prices for gold, Brent crude oil, the NASDAQ composite index, and a broad USD index (de Wolff et al., 2020). Details are as follows:

- **Oil (USD):** Europe Brent spot price from the U.S. energy information administration (EIA).
- **USD Index:** Federal Reserve trade-weighted nominal broad U.S. dollar index (FRED).
- **Gold (USD):** London bullion market association (LBMA) gold price at 10:30 a.m. London time, distributed via FRED.
- **NASDAQ:** Adjusted daily closing prices of the NASDAQ Composite index (symbol ÎXIC, Yahoo Finance).

**EEG dataset:** Electroencephalography (EEG) recordings from human neonates. Multi-channel EEG was recorded from 79 term neonates admitted to the neonatal intensive care unit (NICU) at the

Helsinki University Hospital. The median recording duration was 74 minutes (IQR: 64 to 96 minutes). For our experiments, we use eight of the available twenty-two sensors as outputs: Fp1, Fp2, Fz, Cz, T3, T4, O1, and O2. These correspond to standard electrode positions in the international 10–20 EEG system, covering frontal, central, temporal, and occipital regions. This results in an eight-dimensional multi-output regression task, enabling the study of cross-correlations between EEG channels.

**Currency dataset:** Daily currency exchange rates with respect to the U.S. dollar, collected over the two-year period 2017–2018. We consider ten major currencies in addition to the U.S. dollar reference: Canadian dollar (CAD), Euro (EUR), Japanese yen (JPY), British pound sterling (GBP), Swiss franc (CHF), Australian dollar (AUD), Hong Kong dollar (HKD), New Zealand dollar (NZD), South Korean won (KRW), and Mexican peso (MXN).

**Bramble dataset:** A real-world environmental dataset of tidal height measurements from four coastal weather stations in South England: Bramblemet, Cambermet, Chimet, and Sotonmet. The stations have been continuously recording since April 2012, with measurements every five minutes. For our experiments, we use the tidal height data from a one-week period in June 2020. To improve trainability, the data are detrended before model fitting, and 90% of the data points are randomly removed to accelerate training and reduce memory requirements.

## C.2 EXPERIMENT RESULTS

All datasets are randomly partitioned into 70% training and 30% testing splits. Evaluation is conducted on the test set over five independent runs. Table 7 reports the NMAE results on the GONU dataset, with corresponding regression plots shown in Figures 6–11. Results on the EEG dataset are summarized in Table 8 with regression plots in Figures 12–17. For the currency dataset, NMAE scores are presented in Table 9, 10 and regression plots in Figures 18–23. Finally, Table 11 provides the NMAE results on the Bramble dataset, with regression plots in Figures 24–29.

The proposed MO-LRN kernel consistently outperforms all baselines on the cross-domain multi-output datasets, attaining the lowest overall NMAE and delivering superior or comparable accuracy across individual outputs. The LMC-NGSM kernel typically emerges as the second-best performer. These findings indicate that non-stationary patterns are widespread in time-series data, and capturing them is essential for effective regression. In contrast, stationary kernels such as MOSM, LMC-SM, and CONV show inferior performance, as they lack the ability to capture non-stationary patterns.

Among the non-stationary kernels, both LMC-NGSM and MOHSM fall short compared to our proposed approach, as their spectral formulations inherently restrict expressiveness. In particular, LMC-NGSM constructs kernels as linear combinations of NG-SM components, forcing auto- and cross-covariances to share the same structure. This restriction limits its ability to capture complex cross-output interactions. Although MOHSM is theoretically more flexible than LMC-NGSM, its excessive parameterization makes training difficult, often leading to suboptimal convergence and weaker empirical performance.

## D THE USE OF LARGE LANGUAGE MODELS (LLMs)

We used large language models (LLMs) only for language polishing and readability improvements.

## E FUTURE WORK

In flexible MOGP kernels (Parra and Tobar, 2017; Altamirano and Tobar, 2022), scalability remains an open challenge. One potential direction is the use of inducing point methods, which offer substantial computational savings. However, a key limitation is that they often overlook redundancy from repeated input locations across outputs, leading to inefficient representations and possible numerical instability. Another promising approach is random Fourier features (RFFs). While RFFs provide a straightforward route to linear-time approximations, applying them to complex MOGP kernels poses difficulties, particularly in ensuring the positive definiteness of the joint kernel matrix. In future work, we plan to address these issues to enhance the scalability of flexible MOGP kernels including our MO-LRN and extend their applicability to a broader range of tasks.

**Table 7:** Comparison of NMAE across kernels on four outputs (Oil, Gold, NASDAQ, USD) of GONU dataset. **Mean and standard deviation** are computed over five runs. **Best (lowest)** is bold, **second-best** is underlined.

| MODEL | Oil | Gold | NASDAQ | USD | OVERALL |
|---|---|---|---|---|---|
| METRIC | | | NMAE | | |
| CONV | $0.02333 \pm 0.00035$ | $0.06135 \pm 0.00092$ | $0.03221 \pm 0.00050$ | $0.008239 \pm 0.00012$ | $0.03128 \pm 0.00047$ |
| LMC-SM | $0.02057 \pm 0.00031$ | $0.05301 \pm 0.00080$ | $0.01966 \pm 0.00029$ | $0.009844 \pm 0.00015$ | $0.02577 \pm 0.00039$ |
| MOHSM | $0.04836 \pm 0.00073$ | $0.12455 \pm 0.00187$ | $0.04644 \pm 0.00070$ | $0.020996 \pm 0.00032$ | $0.06059 \pm 0.00091$ |
| MOSM | $\underline{0.01363 \pm 0.00020}$ | $0.05293 \pm 0.00079$ | $0.02515 \pm 0.00038$ | $\underline{0.005669 \pm 0.00009}$ | $0.02485 \pm 0.00037$ |
| LMC-NGSM | $0.01747 \pm 0.00026$ | $\underline{0.02519 \pm 0.00038}$ | $\underline{0.01559 \pm 0.00023}$ | $0.005906 \pm 0.00009$ | $\underline{0.01654 \pm 0.00024}$ |
| MO-LRN | $\mathbf{0.01218 \pm 0.00023}$ | $\mathbf{0.02513 \pm 0.00195}$ | $\mathbf{0.01439 \pm 0.00278}$ | $\mathbf{0.004996 \pm 0.000632}$ | $\mathbf{0.01467 \pm 0.00723}$ |

**Table 8:** Comparison of **NMAE** across kernels on the eight outputs (Fp1, Fp2, Fz, Cz, T3, T4, O1, O2) of EEG dataset. **Mean and standard deviation** over five runs. **Best (lowest)** in bold; **second-best** underlined.

| MODEL | Fp1 | Fp2 | Fz | Cz | T3 | T4 | O1 | O2 | OVERALL |
|---|---|---|---|---|---|---|---|---|---|
| METRIC | | | | | NMAE | | | | |
| CONV | $0.831 \pm 0.005$ | $0.885 \pm 0.007$ | $0.843 \pm 0.013$ | $0.882 \pm 0.005$ | $0.694 \pm 0.003$ | $0.742 \pm 0.003$ | $0.944 \pm 0.009$ | $0.736 \pm 0.006$ | $0.820 \pm 0.006$ |
| LMC-SM | $0.846 \pm 0.006$ | $0.876 \pm 0.007$ | $0.777 \pm 0.013$ | $0.986 \pm 0.005$ | $0.593 \pm 0.003$ | $0.704 \pm 0.003$ | $0.928 \pm 0.009$ | $0.613 \pm 0.006$ | $0.791 \pm 0.006$ |
| MOHSM | $0.862 \pm 0.006$ | $0.820 \pm 0.006$ | $0.812 \pm 0.013$ | $0.702 \pm 0.005$ | $0.689 \pm 0.003$ | $0.670 \pm 0.003$ | $\underline{0.735 \pm 0.008}$ | $0.670 \pm 0.005$ | $0.745 \pm 0.007$ |
| MOSM | $0.876 \pm 0.006$ | $0.833 \pm 0.007$ | $0.770 \pm 0.013$ | $\underline{0.610 \pm 0.005}$ | $0.702 \pm 0.003$ | $0.654 \pm 0.003$ | $0.750 \pm 0.008$ | $0.652 \pm 0.005$ | $0.731 \pm 0.007$ |
| LMC-NGSM | $\underline{0.548 \pm 0.004}$ | $\underline{0.551 \pm 0.004}$ | $0.714 \pm 0.010$ | $1.313 \pm 0.009$ | $\mathbf{0.390 \pm 0.003}$ | $\mathbf{0.302 \pm 0.003}$ | $0.803 \pm 0.008$ | $0.267 \pm 0.003$ | $0.611 \pm 0.006$ |
| **MO-LRN** | $\mathbf{0.502 \pm 0.004}$ | $\mathbf{0.503 \pm 0.004}$ | $\mathbf{0.579 \pm 0.009}$ | $\mathbf{0.468 \pm 0.004}$ | $\underline{0.527 \pm 0.004}$ | $\underline{0.354 \pm 0.003}$ | $\mathbf{0.517 \pm 0.005}$ | $\mathbf{0.259 \pm 0.003}$ | $\mathbf{0.464 \pm 0.005}$ |

**Table 9:** Comparison of **NMAE** across kernels on five currency outputs (EUR, CAD, JPY, GBP, CHF). **Mean and standard deviation** over five runs. **Best (lowest)** in bold; **second-best** underlined.

| MODEL | EUR/USD | CAD/USD | JPY/USD | GBP/USD | CHF/USD |
|---|---|---|---|---|---|
| METRIC | | | NMAE | | |
| CONV | $0.00563 \pm 0.00017$ | $\underline{0.00683 \pm 0.00020}$ | $\underline{0.00721 \pm 0.00022}$ | $\underline{0.00770 \pm 0.00023}$ | $0.00593 \pm 0.00018$ |
| LMC-SM | $0.00653 \pm 0.00020$ | $0.00857 \pm 0.00026$ | $0.00742 \pm 0.00022$ | $\underline{0.00732 \pm 0.00022}$ | $0.00702 \pm 0.00021$ |
| MOHSM | $0.29028 \pm 0.00871$ | $3.51393 \pm 0.10542$ | $0.01158 \pm 0.00035$ | $0.07513 \pm 0.00225$ | $0.08108 \pm 0.00243$ |
| MOSM | $\underline{0.00411 \pm 0.00012}$ | $0.01326 \pm 0.00040$ | $0.01415 \pm 0.00042$ | $0.00917 \pm 0.00028$ | $0.00629 \pm 0.00019$ |
| LMC-NGSM | $0.00548 \pm 0.00016$ | $\mathbf{0.00445 \pm 0.00013}$ | $\mathbf{0.00713 \pm 0.00021}$ | $0.00921 \pm 0.00028$ | $\underline{0.00386 \pm 0.00012}$ |
| **MO-LRN** | $\mathbf{0.00239 \pm 0.00007}$ | $0.00989 \pm 0.00030$ | $0.00923 \pm 0.00028$ | $\mathbf{0.00731 \pm 0.00022}$ | $\mathbf{0.00221 \pm 0.00007}$ |

**Table 10:** Comparison of **NMAE** across kernels on five currency outputs (AUD, HKD, NZD, KRW, MXN) plus OVERALL. **Mean and standard deviation** are computed over five runs. **Best (lowest)** in bold; **second-best** underlined.

| MODEL | AUD/USD | HKD/USD | NZD/USD | KRW/USD | MXN/USD | OVERALL |
|---|---|---|---|---|---|---|
| METRIC | | | | NMAE | | |
| CONV | $0.00672 \pm 0.00020$ | $0.00036 \pm 0.00001$ | $0.00669 \pm 0.00020$ | $\underline{0.00635 \pm 0.00019}$ | $0.01344 \pm 0.00040$ | $0.00669 \pm 0.00007$ |
| LMC-SM | $0.00733 \pm 0.00022$ | $0.00136 \pm 0.00004$ | $0.00836 \pm 0.00025$ | $0.00834 \pm 0.00025$ | $\underline{0.00525 \pm 0.00016}$ | $0.00675 \pm 0.00007$ |
| MOHSM | $0.77411 \pm 0.02322$ | $0.00237 \pm 0.00007$ | $0.01992 \pm 0.00060$ | $0.01785 \pm 0.00054$ | $0.02673 \pm 0.00080$ | $0.48130 \pm 0.00472$ |
| MOSM | $\underline{0.00530 \pm 0.00016}$ | $0.00026 \pm 0.00001$ | $\underline{0.00595 \pm 0.00018}$ | $\mathbf{0.00304 \pm 0.00009}$ | $0.00686 \pm 0.00021$ | $0.00684 \pm 0.00007$ |
| LMC-NGSM | $0.00543 \pm 0.00016$ | $0.00034 \pm 0.00001$ | $0.00812 \pm 0.00024$ | $0.00774 \pm 0.00023$ | $0.00685 \pm 0.00021$ | $\underline{0.00586 \pm 0.00006}$ |
| **MO-LRN** | $\mathbf{0.00513 \pm 0.00024}$ | $\mathbf{0.00012 \pm 0.00003}$ | $\mathbf{0.00326 \pm 0.00010}$ | $0.01084 \pm 0.00033$ | $\mathbf{0.00515 \pm 0.00015}$ | $\mathbf{0.00555 \pm 0.00006}$ |

**Table 11:** Comparison of NMAE across kernels on the four outputs (Bramble, Camber, Chi, Soton) of bramble dataset. **Mean and standard deviation** are computed over five runs, **the best (lowest) value is bold**, and **the second-best is underlined**.

| MODEL | Bramble | Camber | Chi | Soton | OVERALL |
|---|---|---|---|---|---|
| METRIC | | | NMAE | | |
| CONV | $0.0825 \pm 0.0025$ | $0.0621 \pm 0.0019$ | $0.0599 \pm 0.0018$ | $0.1071 \pm 0.0033$ | $0.0779 \pm 0.0024$ |
| LMC-SM | $0.0791 \pm 0.0024$ | $0.0651 \pm 0.0020$ | $\underline{0.0590 \pm 0.0018}$ | $0.1055 \pm 0.0032$ | $0.0772 \pm 0.0024$ |
| MOHSM | $211.530 \pm 2.115$ | $4.027 \pm 0.040$ | $0.609 \pm 0.006$ | $1.269 \pm 0.013$ | $54.858 \pm 0.556$ |
| MOSM | $0.0831 \pm 0.0025$ | $0.0617 \pm 0.0019$ | $0.0594 \pm 0.0018$ | $0.1082 \pm 0.0032$ | $0.0781 \pm 0.0024$ |
| LMC-NGSM | $\underline{0.0504 \pm 0.0015}$ | $\underline{0.0612 \pm 0.0018}$ | $0.0690 \pm 0.0021$ | $\underline{0.0642 \pm 0.0019}$ | $\underline{0.0612 \pm 0.0018}$ |
| **MO-LRN** | $\mathbf{0.0497 \pm 0.0015}$ | $\mathbf{0.00977 \pm 0.00029}$ | $\mathbf{0.0150 \pm 0.00045}$ | $\mathbf{0.0183 \pm 0.00055}$ | $\mathbf{0.0237 \pm 0.00071}$ |

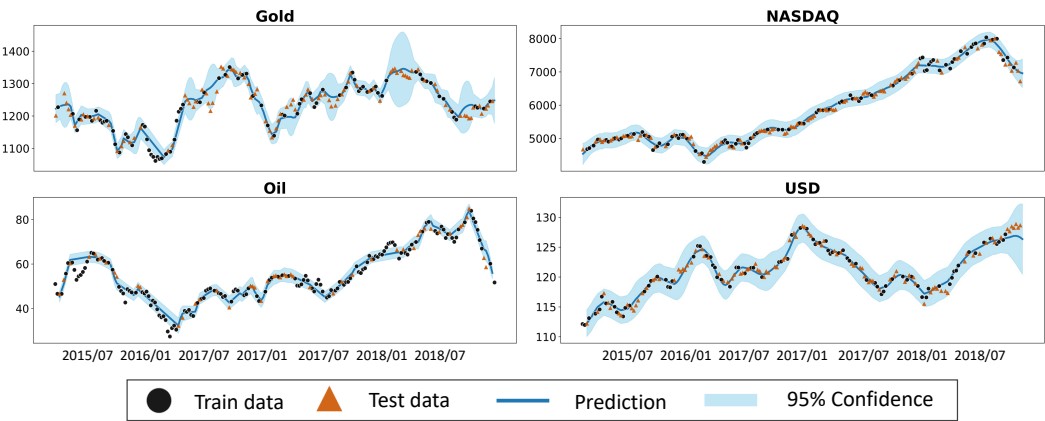

**Figure 6:** MOGP regression plots on the GONU dataset with the MO-LRN kernel.

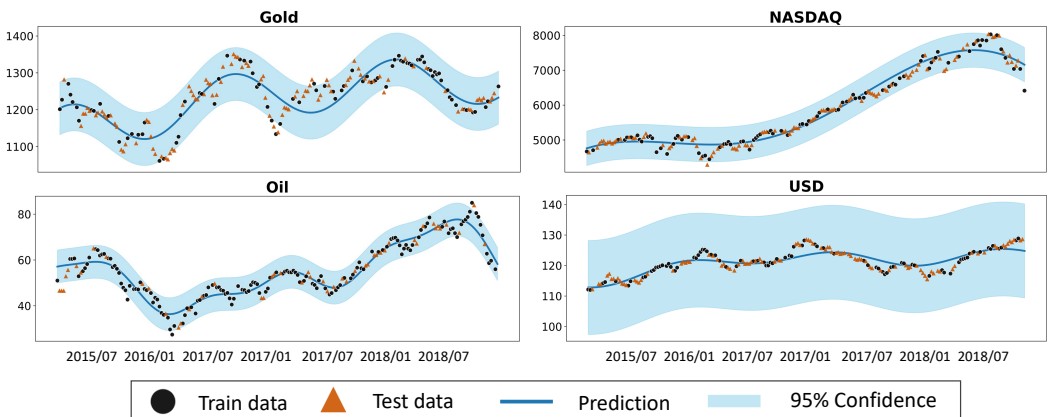

**Figure 7:** MOGP regression plots on the GONU dataset with the MOHSM kernel.

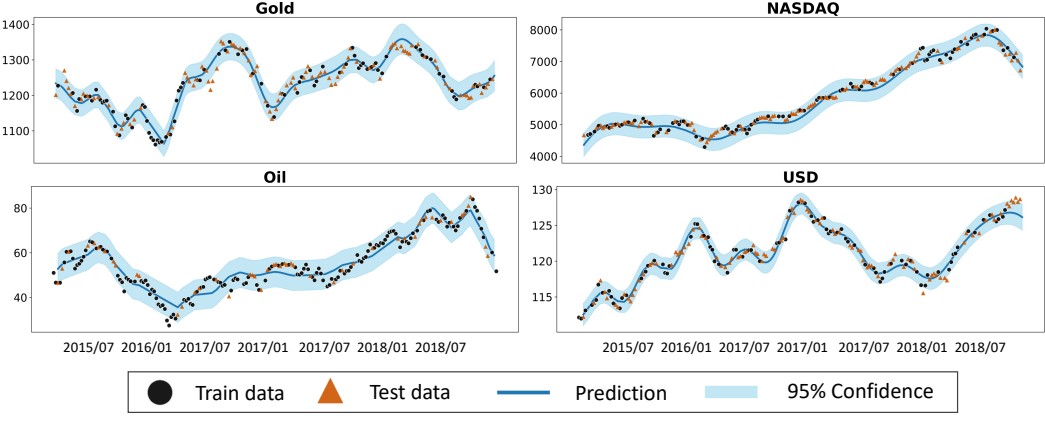

**Figure 8:** MOGP regression plots on the GONU dataset with the MOSM kernel.

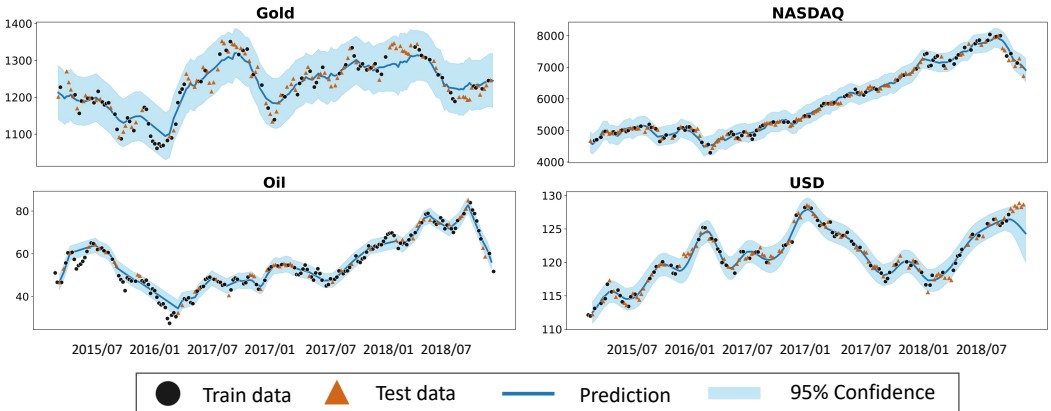

**Figure 9:** MOGP regression plots on the GONU dataset with the LMC-NGSM kernel.

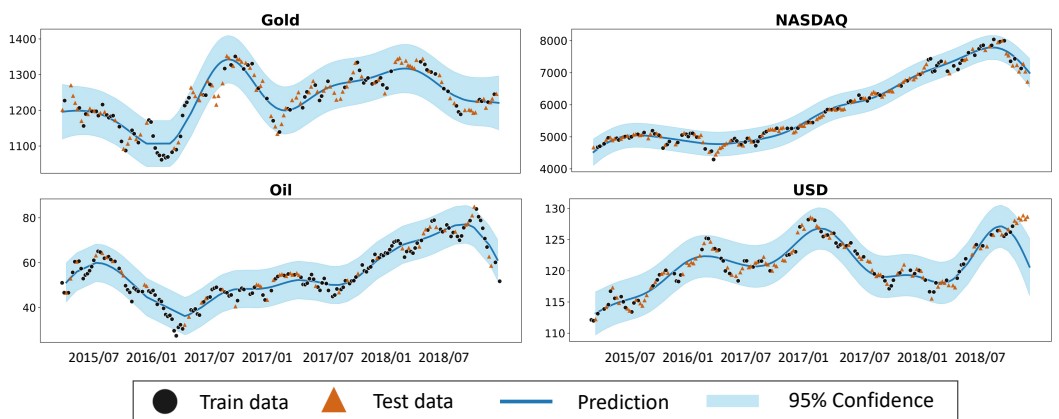

**Figure 10:** MOGP regression plots on the GONU dataset with the LMC-SM kernel.

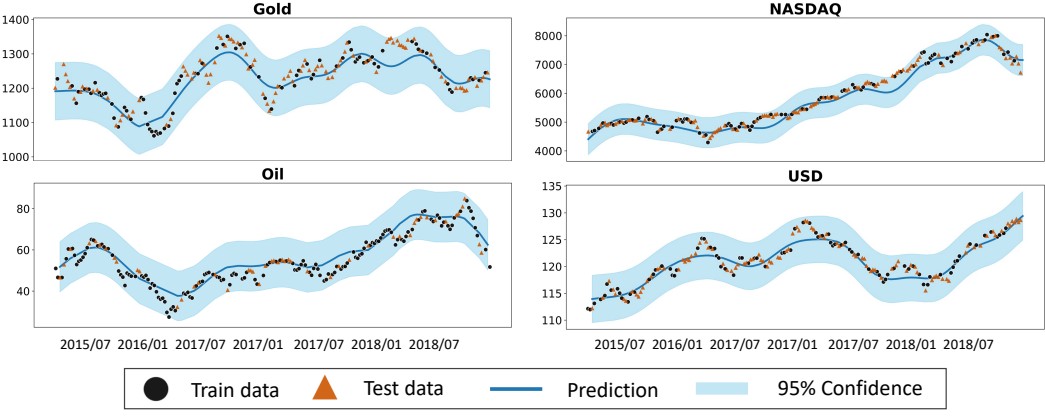

**Figure 11:** MOGP regression plots on the GONU dataset with the CONV kernel.

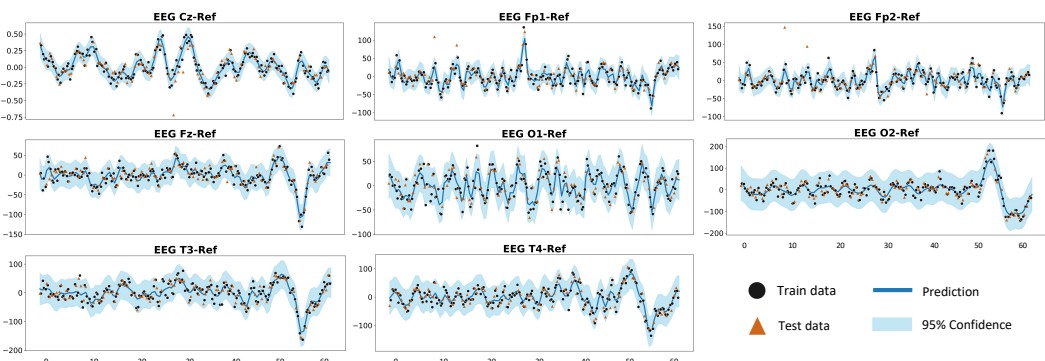

**Figure 12:** MOGP regression plots on the EEG dataset with the MO-LRN kernel.

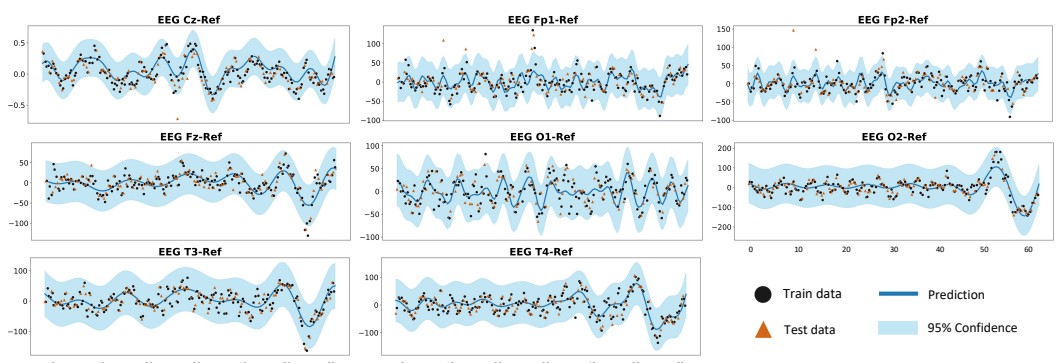

**Figure 13:** MOGP regression plots on the EEG dataset with the MOHSM kernel.

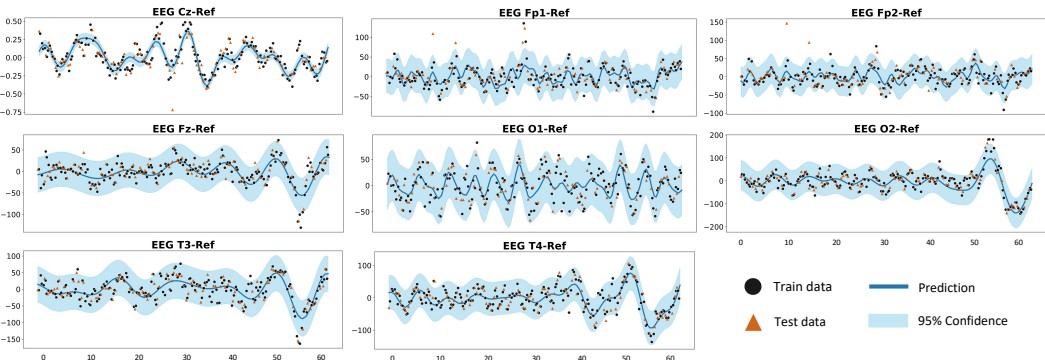

**Figure 14:** MOGP regression plots on the EEG dataset with the MOSM kernel.

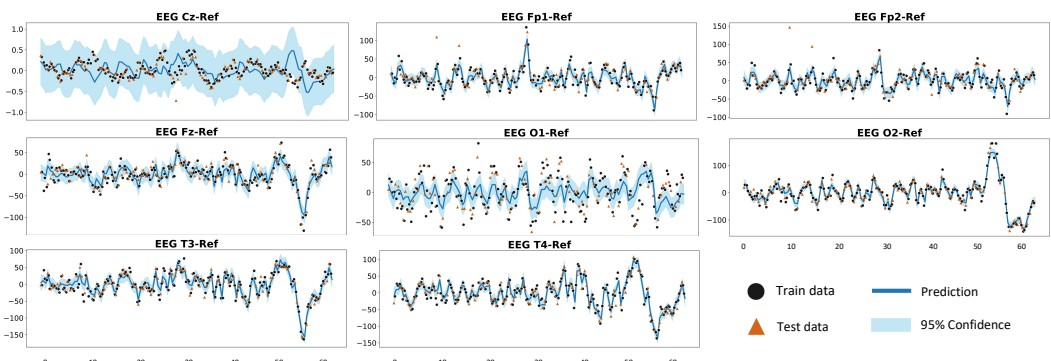

**Figure 15:** MOGP regression plots on the EEG dataset with the LMC-NGSM kernel.

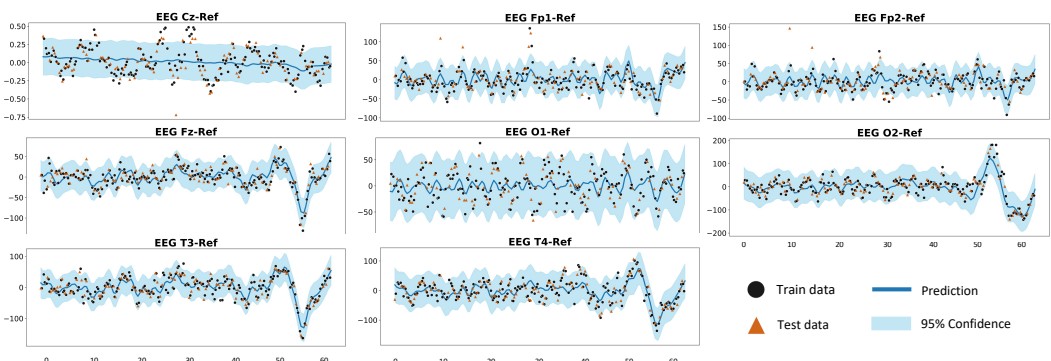

**Figure 16:** MOGP regression plots on the EEG dataset with the LMC-SM kernel.

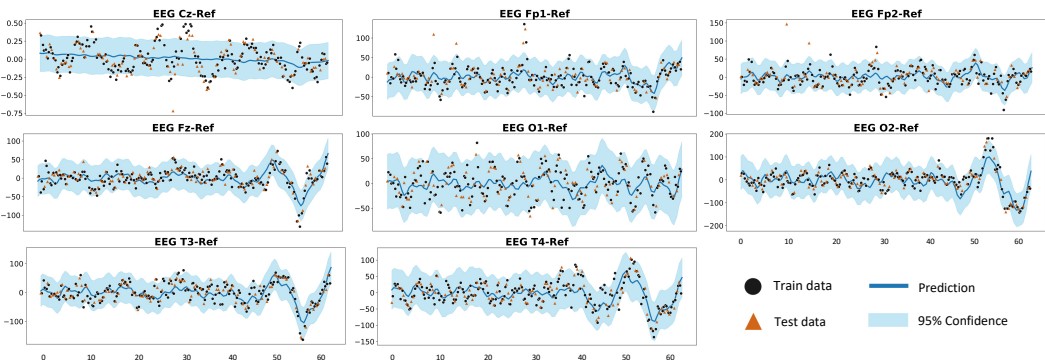

**Figure 17:** MOGP regression plots on the EEG dataset with the CONV kernel.

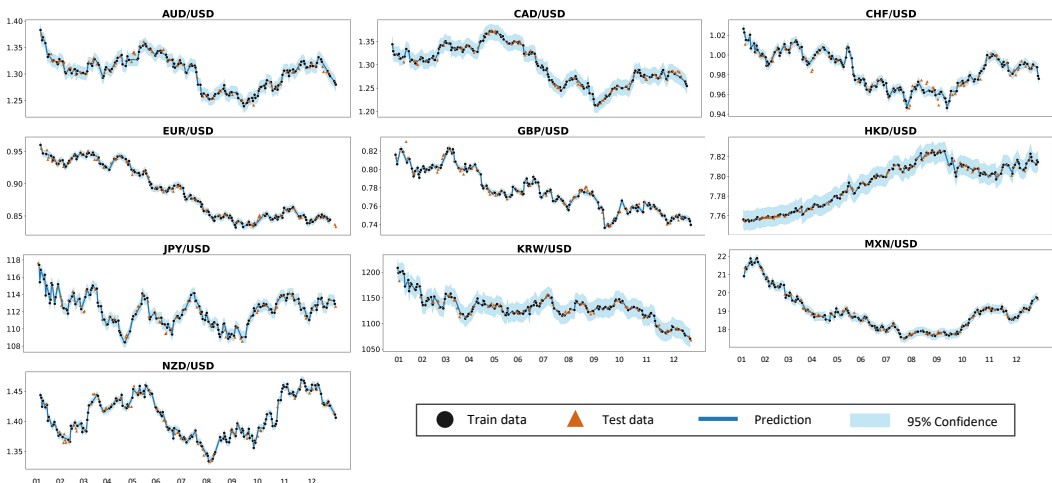

**Figure 18:** MOGP regression plots on the currency dataset with the MO-LRN kernel.

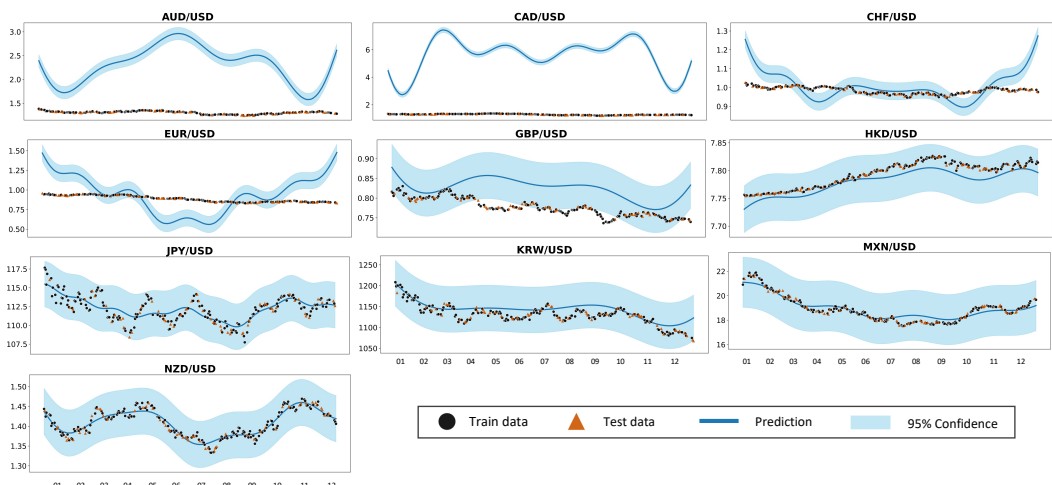

**Figure 19:** MOGP regression plots on the currency dataset with the MOHSM kernel.

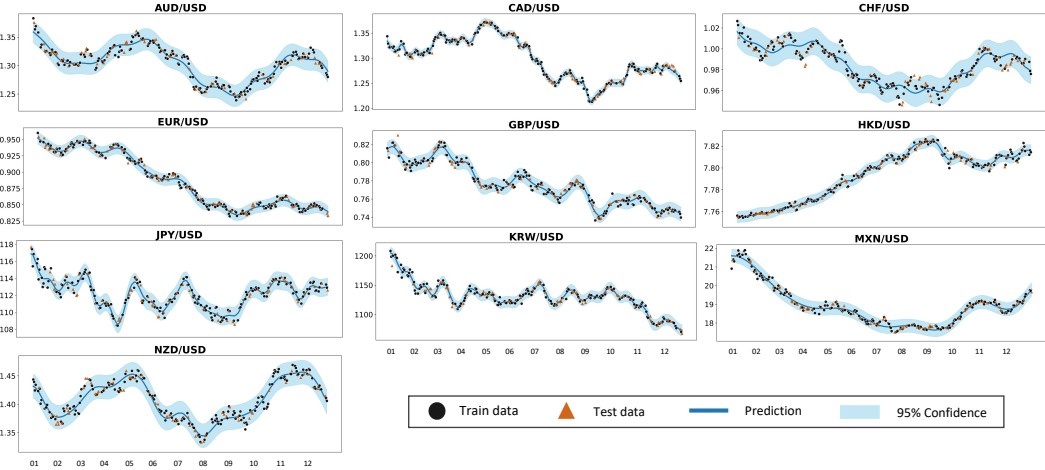

**Figure 20:** MOGP regression plots on the currency dataset with the MOSM kernel.

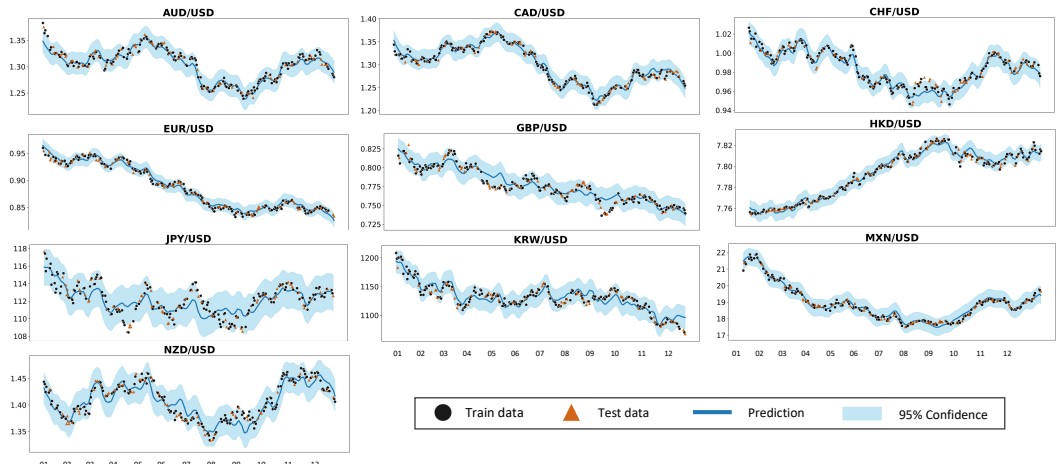

**Figure 21:** MOGP regression plots on the currency dataset with the LMC-NGSM kernel.

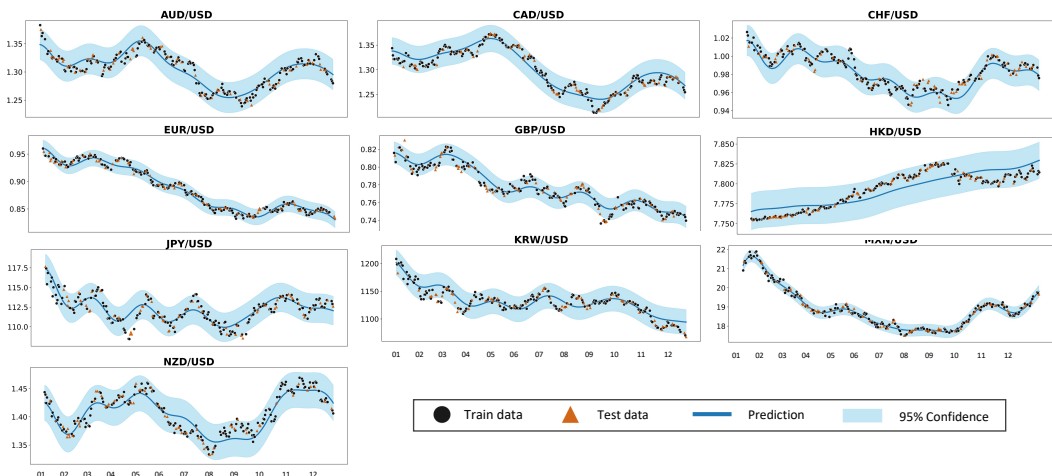

**Figure 22:** MOGP regression plots on the currency dataset with the LMC-SM kernel.

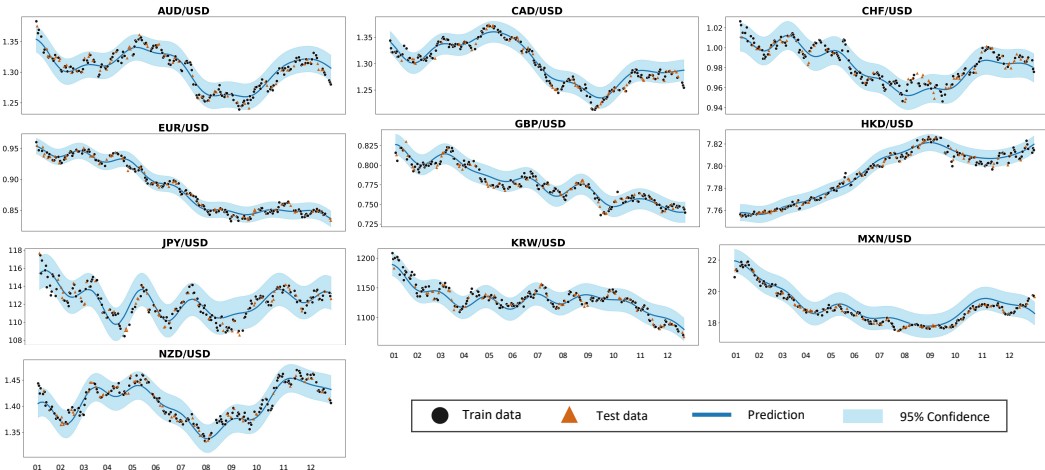

**Figure 23:** MOGP regression plots on the currency dataset with the CONV kernel.

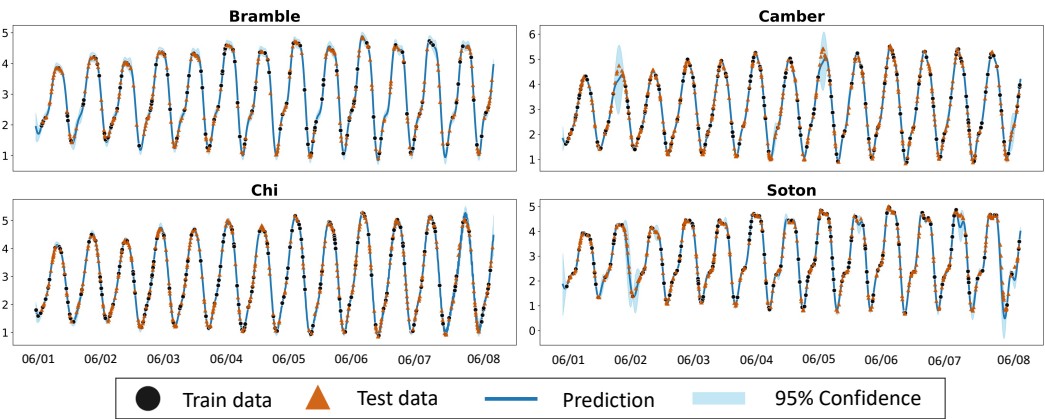

Figure 24: MOGP regression plots on the bramble dataset with the MO-LRN kernel.

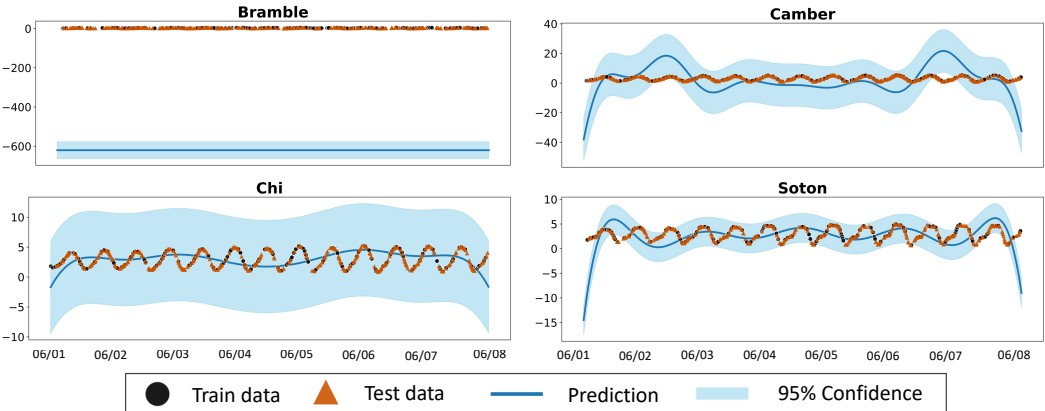

Figure 25: MOGP regression plots on the bramble dataset with the MOHSM kernel.

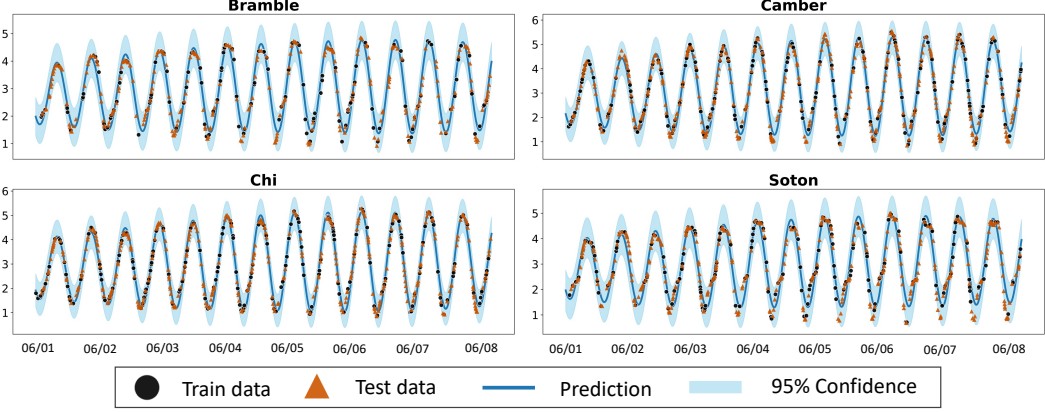

Figure 26: MOGP regression plots on the bramble dataset with the MOSM kernel.

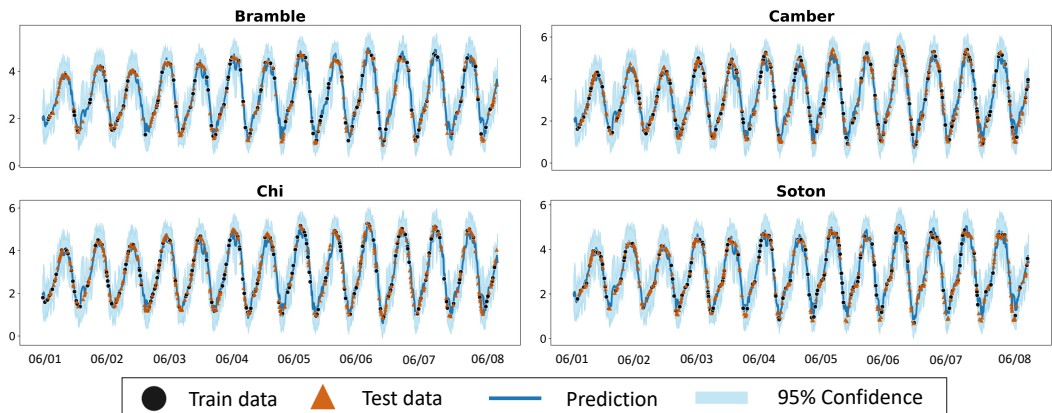

**Figure 27:** MOGP regression plots on the bramble dataset with the LMC-NGSM kernel.

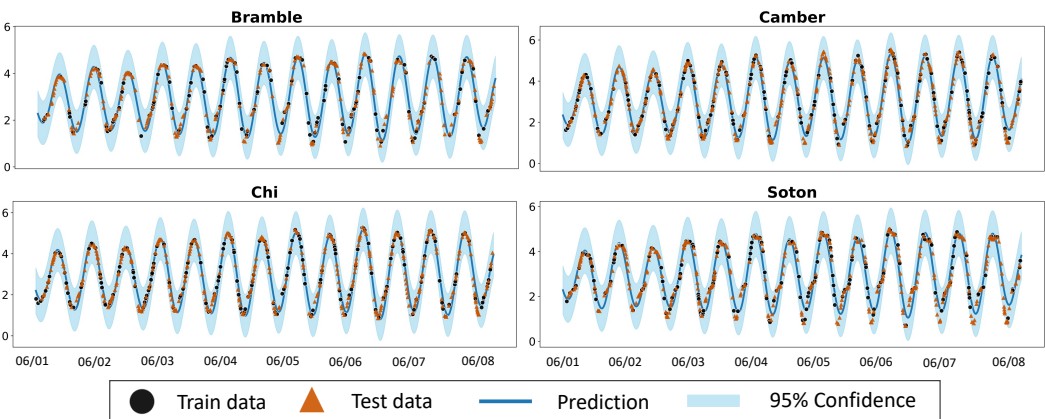

**Figure 28:** MOGP regression plots on the bramble dataset with the LMC-SM kernel.

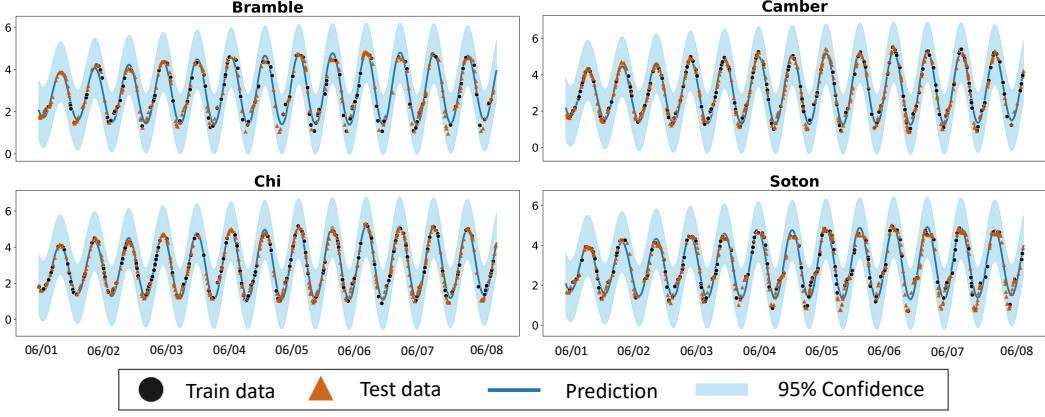

**Figure 29:** MOGP regression plots on the bramble dataset with the CONV kernel.

