# OpenReview forum: "Revisiting Nonstationary Kernel Design for Multi-Output Gaussian Processes"
_ICLR.cc/2026/Conference — ICLR 2026 Poster_

### Official Review · Reviewer_ZUMe · 2025-10-27

**Soundness:** 4
**Presentation:** 3
**Contribution:** 4
**Rating:** 10
**Confidence:** 4

**Summary:**

I applaud the authors for revisiting multi-task GPs through non-stationary kernels. I fully agree with the paper's motivation. The manuscript is well written and a delightful, helpful read. My comments and questions mainly seek clarification.

**Strengths:**

- The paper proposes a new, very flexible kernel for multi-task GPs, which is an important area of research
- The manuscript is well-written and comprehensible without too much effort.
- Sound theory.
- A reasonable number of test cases.
- A good choice of competitor methods.

**Weaknesses:**

- Some extra implementation details would be good. I found it hard to reproduce the kernel from Definition 1. For instance, one could describe typical bounds for the hyperparameter.

**Questions:**

- Are there restrictions or bounds on the m_ij, z_ij, S_ij^(q), and other hyperparameters?
- What is the reasoning behind the choice of competitor methods?
- Why is there no comparison to a baseline coregionalization?

---

> ### Author Response · Authors · 2025-11-20
> **Response to Reviewer ZUMe**
>
> Thanks for your insightful feedback and comments. We sincerely appreciate the time and effort you invested in reviewing our work.
>
> >**C1: Some extra implementation details would be good. I found it hard to reproduce the kernel from Definition 1. For instance, one could describe typical bounds for the hyperparameter.**
>
> >**C2: Are there restrictions or bounds on the $m_{ij}$, $z_{ij}$, $S_{ij}^{(q)}$, and other hyperparameters?**
>
> **Response:** We thank the reviewer for pointing this out, and we apologize for the difficulty in reproducing the kernel directly from Definition 1. To improve reproducibility, we have implemented our kernel following the interface conventions of the MOGP Toolkit (MOGPTK), and the corresponding code and detailed documentation will be released in the camera-ready version on GitHub.
>
> To further enhance clarity and help readers reconstruct the kernel, we have added more illustrations and explanations directly in Definition 1. Specifically, the kernel is fully determined by
>
> $$
> \boldsymbol{\theta} = \\{ w_k^{(q)},\\, \boldsymbol{\mu_{k1}^{(q)}},\\, \boldsymbol{\mu_{k2}^{(q)}},\\, \boldsymbol{\sigma_{k1}^{(q)}},\\, \boldsymbol{\sigma_{k2}^{(q)}},\\, \rho_k^{(q)} \\}_{k \in \\{i,j\\},\, q=1,\dots,Q}
> $$
>
> and all parameters appearing in Definition 1 follow from straightforward algebraic transformations of these quantities (the explicit formulas are provided in Definition 2 in Appendix A.2).
>
> Regarding the bounds for the hyperparameters, the base hyperparameters $\boldsymbol{\theta}$ themselves do not require any manual constraints; they are simply randomly initialized and then fully learned during training.
>
> >**C3: What is the reasoning behind the choice of competitor methods?**
>
> **Response:** We choose representative stationary kernels and non-stationary kernels to ensure a comprehensive comparison. The specific reasons are as follows.
>
> #### **1. Stationary kernels**
> To highlight the necessity of non-stationary designs, we include several classical stationary MOGP kernels (e.g., **MOSM**, **LMC-SM**, **CONV**) which cannot model input-dependent variations both empirically and theoretically.
>
> #### **2. Non-stationary kernels**
> First, since our work revisits and analyzes the limitations of **MOHSM**, it is essential to include it as a baseline.  Second, any LMC combined with a non-stationary single-output kernel becomes a valid non-stationary MOGP kernel. To cover this class, we adopt **NG-SM**, the most expressive available single-output spectral kernel, as the base kernel which ensures that this family is represented by its strongest member.
>
> >**C4: Why is there no comparison to a baseline coregionalization?**
>
> **Response:** In fact, we have compared our model against several coregionalization-based baselines, including LMC-NGSM and LMC-SM, all of which follow the linear model of coregionalization (LMC) framework. The detailed descriptions can be found in Appendix B.2.1.

---

### Official Review · Reviewer_Ls9o · 2025-10-30

**Soundness:** 3
**Presentation:** 3
**Contribution:** 3
**Rating:** 8
**Confidence:** 3

**Summary:**

The paper introduces a multi-output version of the next-gen SM kernel from Yang (2025).

**Strengths:**

* The proposed kernel gives an elegant solution to the problem.
* The method has good results.

**Weaknesses:**

* The appendix contains the hyperparameter values, but only for MO-LRN, not for the other kernels.
  * The appendix also specifies how the kernel was optimized for MO-LRN (500 iterations of Adam), but not for the other kernels. Is the same procedure used for all kernels?
  * It is not clear how the hyperparameters like learning rate were chosen.
* The source code is not released (yet)

**Questions:**

* Table 2: "$I$ denotes the index in the LMC summation over latent processes, with $I < V$."
   Should this be $I \le V$?
 * Table 2: what does "Expressive kernel" mean?
 * Eq (1) is missing parentheses around the 4 terms in the integral.
 * "When ω_1 = ω_2, this theorem reduces to Bochner’s theorem."
    ω are not parameters of the theorem, so this statement makes no sense. Do you mean to say that the theorem reduces to Bochner's theorem when u(ω_1,ω_2)=δ(ω_1-ω_2)u(ω), so when u is non-zero only when ω_1 = ω_2?

---

> ### Author Response · Authors · 2025-11-20
> **Response to reviewer Ls9o**
>
> We thank the reviewer for the invaluable feedback and comments. Below are our detailed responses.
>
> >**C1:The appendix contains the hyperparameter values, but only for MO-LRN, not for the other kernels. The appendix also specifies how the kernel was optimized for MO-LRN (500 iterations of Adam), but not for the other kernels.**
> >* **Is the same procedure used for all kernels?**
> >* **It is not clear how the hyperparameters like learning rate were chosen.**
>
> **Response:** We have updated **Appendix B.3** to include the full hyperparameter tables for all baselines (Tables 5 and 6), including the number of optimization iterations. We choose a learning rate that ensures stable optimization behavior for MOHSM.
>
> >**C2:The source code is not released (yet)**
>
> **Response:** At this stage, we have only uploaded a Jupyter notebook containing the toy example simulation to support basic reproducibility. The full source code will be publicly released in the camera-ready version.
>
> >**C3:Here, $I$ denotes the index in the LMC summation over latent processes, with $I < V$. Should this be $I <= V$ ?**
>
> **Response:** We have corrected this in Table 2 in the revised version.
>
> >**C4:Table 2: what does "Expressive kernel" mean?**
>
> **Response:** By “expressive kernel,” we mean that its parametrization in the dual (spectral) space can approximate a broad class of spectral densities, allowing the kernel to capture rich covariance structures.
>
> >**C5: Eq (1) is missing parentheses around the 4 terms in the integral.**
>
> **Response:** Done.
>
> >**C6:"When $\omega_1 = \omega_2$, this theorem reduces to Bochner’s theorem." $\omega$ are not parameters of the theorem, so this statement makes no sense. Do you mean to say that the theorem reduces to Bochner’s theorem when  $u(\omega_1, \omega_2) = \delta(\omega_1 - \omega_2) u(\omega)$, so when $u$ is non-zero only when $\omega_1 = \omega_2$?**
>
> **Response:** We sincerely thank the reviewer for the careful and rigorous observation. We revised the description in Theorem 1 to:
>
> *When $u(\omega_1,\omega_2)=\delta(\omega_1-\omega_2)\,u(\omega)$ (with $\omega \triangleq \omega_1 = \omega_2$),
> this theorem reduces to Bochner’s theorem.*

---

### Official Review · Reviewer_kt5w · 2025-11-01

**Soundness:** 3
**Presentation:** 3
**Contribution:** 3
**Rating:** 4
**Confidence:** 4

**Summary:**

The manuscript presents a novel formulation of a multi-output non-stationary spectral kernel. A low rank approximation is adopted in order to maintain a manageable number of trainable parameters. The model is benchmarked against synthetic and real world data, and shown to be competitive for point prediction tasks. The approach also demonstrates favourable computational efficiency.

**Strengths:**

The manuscript is well written and clearly presented

The approach is a novel solution to tackling a challenging problem in the literature.

The experimental results are clearly presented along with a suitably broad selection of benchmarks and a good set of baselines.

**Weaknesses:**

My primary concern with this work is the absence of quantitative probabilistic performance metrics. The metrics shown, such as MAE and RMSE, test only the quality of the point predictions of the model. But here we are working with probabilistic models, so without metrics such as NLPD to verify the uncertainty calibration, it is difficult to recommend acceptance. This concern would hold for any probabilistic model, but is particularly acute here where we have many degrees of freedom (due to non-stationarity and multi-output), as it becomes more challenging to regulate the full posterior.

It is unclear what we are sacrificing when moving to the low rank setting.

Scalability remains a significant concern that is only briefly discussed in the Appendix. For multi-output datasets that are long enough to reveal detectable non-stationary features, we quickly hit the upper limits of Figure 4, where the computational cost becomes prohibitive.

**Questions:**

Under what conditions should we expect this formalism to fail? It might be helpful to construct a somewhat adversarial synthetic example where we know the model cannot reproduce the cross-covariances, and see how it performs.

As pointed out in 4.1, spectral kernels tend to be highly sensitive to the initialisation strategy, but it was not clear to me what initialisation is used for MO-LRN? And how should practitioners decide on a suitable configuration such as Q for a given dataset?

---

> ### Author Response · Authors · 2025-11-20
> **Response to Reviewer kt5w (1/2)**
>
> We acknowledge and appreciate the reviewer’s detailed feedback. Please see our responses addressing each comment below.
>
> >**C1: My primary concern with this work is the absence of quantitative probabilistic performance metrics. The metrics shown, such as MAE and RMSE, test only the quality of the point predictions of the model. But here we are working with probabilistic models, so without metrics such as NLPD to verify the uncertainty calibration, it is difficult to recommend acceptance. This concern would hold for any probabilistic model, but is particularly acute here where we have many degrees of freedom (due to non-stationarity and multi-output), as it becomes more challenging to regulate the full posterior.**
>
> **Response:** We appreciate the reviewer’s insightful suggestion. As recommended, we have added NLPD to quantify uncertainty and calibration in Section 4.2 of the revision. For convenience, we also summarize the newly added results in the table below.
>
> | TARGET | OT | HUFL | HULL | MUFL | MULL | LUFL | LULL | OVERALL |
> | :--- | :--- | :--- | :--- | :--- | :--- | :--- | :--- | :--- |
> | CONV | 0.384 ± 0.045 | 0.484 ± 0.034 | 0.156 ± 0.060 | 0.647 ± 0.147 | 1.328 ± 0.023 | 0.877 ± 0.069 | 0.748 ± 0.320 | 0.661 ± 0.069 |
> | LMC-SM | 0.286 ± 0.021 | 0.429 ± 0.009 | 0.066 ± 0.012 | 0.554 ± 0.008 | 1.283 ± 0.033 | 0.889 ± 0.002 | 0.506 ± 0.198 | 0.573 ± 0.031 |
> | MOHSM | 0.733 ± 0.022 | 0.757 ± 0.019 | 0.702 ± 0.023 | 0.809 ± 0.016 | 0.920 ± 0.011 | 0.881 ± 0.011 | 0.740 ± 0.021 | 0.792 ± 0.018 |
> | MOSM | 0.473 ± 0.073 | 0.467 ± 0.118 | 0.208 ± 0.118 | 0.586 ± 0.116 | 1.048 ± 0.038 | 0.846 ± 0.047 | 0.103 ± 0.075 | 0.533 ± 0.043 |
> | LMC-NGSM | 0.146 ± 0.167| 0.365 ± 0.046| -0.019 ± 0.133 | **0.501 ± 0.020** | 0.866 ± 0.367 | 0.771 ± 0.049 | 0.042 ± 0.172| 0.382 ± 0.133|
> | **MO-LRN** | **-0.159 ± 0.012** | **0.334 ± 0.087** | **-0.271 ± 0.038** | 0.505 ± 0.097 | **0.691 ± 0.092** | **0.385 ± 0.007** | **-0.323 ± 0.014** | **0.166 ± 0.031** |
>
> From the table, we observe that our method consistently achieves a lower NLPD compared to all of the baseline methods. This indicates that our model not only provides accurate point estimates, but also produces well-calibrated uncertainty estimates.
>
> >**C2: It is unclear what we are sacrificing when moving to the low rank setting.**
>
> >**C4: Under what conditions should we expect this formalism to fail? It might be helpful to construct a somewhat adversarial synthetic example where we know the model cannot reproduce the cross-covariances, and see how it performs.**
>
> **Response:** We thank the reviewers for these helpful questions. As clarified in **Remark 2**, only the diagonal spectra, $P_{ii}(\boldsymbol{\omega})$, are independently parameterized and therefore dense. The cross-spectra, $P_{ij}(\boldsymbol{\omega})$, in contrast, are generated as inner products of the same set of Gaussian vectors. Consequently, all of the off-diagonal terms will lie in the span of these shared components and cannot vary independently. This is the main tradeoff introduced by the low-rank form. Empirically, we observe that this restriction does not affect performance on the datasets considered.
>
> A simple adversarial example is to construct a process whose spectral density matrix has rank strictly larger than the number of shared components $Q$. Such a high-rank structure is easy to construct synthetically, but most classical multi-output kernels (e.g., LMC, ICM, CONV) are inherently low-rank spectral formulations and are well known to perform competitively in practice, even though their induced spectral spaces remain limited in flexibility [1].
>
> [1] Alvarez, Mauricio A., Lorenzo Rosasco, and Neil D. Lawrence. "Kernels for vector-valued functions: A review." Foundations and Trends® in Machine Learning 4.3 (2012): 195-266.
>
> >**C3: Scalability remains a significant concern that is only briefly discussed in the Appendix. For multi-output datasets that are long enough to reveal detectable non-stationary features, we quickly hit the upper limits of Figure 4, where the computational cost becomes prohibitive.**
>
> **Response:** Thank you for pointing this out. We agree that the runtime analysis is better positioned in the main text, and we have moved it to Section 4 accordingly. Importantly, the curve in Figure 3 (originally called Figure 4) shows that the method that MOHSM quickly hits the upper limit, whose computational cost grows steeply with the number of data points. In contrast, our method remains competitive with all other baselines and does not exhibit a rapid explosion in computational cost. In the revision, we have also made our method more visually prominent in the runtime plot to avoid any misinterpretation.

---

> ### Author Response · Authors · 2025-11-20
> **Response to Reviewer kt5w (2/2)**
>
> > **C5:As pointed out in 4.1, spectral kernels tend to be highly sensitive to the initialisation strategy, but it was not clear to me what initialisation is used for MO-LRN? And how should practitioners decide on a suitable configuration such as Q for a given dataset?**
>
> **Response:** Unlike MOHSM, whose over-parameterization makes it highly sensitive to initialization and often requires carefully designed strategies, our MO-LRN kernel yields stable results under simple random initialization. Specifically, we initialize
> $$
> w_k^{(q)} \sim \boldsymbol{Uniform} (0,1),\quad
> \boldsymbol{\mu_{k1/2}^{(q)}} \sim \mathcal{N}(\boldsymbol{0},\mathbf{I_D}),\quad
> \boldsymbol{\sigma_{k1/2}^{(q)}} = \mathbf{1_D},\quad
> \rho_k^{(q)} = 0.1,
> $$
> where $\mathbf{I_D}$ is the $D \times D$ identity matrix and $\mathbf{1_D}$ is the D-dimensional all-ones vector, and observe consistent performance across random seeds.
>
> For clarity regarding the choice of $Q$, we include an additional experiment in Appendix B.3. Specifically, we report the NMAE and runtime of our model under different values of $Q$, with the results visualized in Figure 5. For convenience, we also provide a table summarizing these metrics.
>
> | Metric       |   1     |   2     |   3     |   4        |   5     |   6     |   7       |   8     |   9     |   10    |
> |--------------|---------|---------|---------|------------|---------|---------|-----------|---------|---------|---------|
> | **NMAE**     | 0.0193  | 0.0183  | 0.0160  | **0.0148** | 0.0151  | 0.0150  | 0.0142    | 0.0140  | 0.0139  | 0.0138  |
> | **Time (s)** | 112.42  | 195.80  | 320.74  | **424.33** | 502.44  | 517.59  | 594.50    | 675.92  | 750.34  | 1002.78 |
>
> As shown in the table, increasing $Q$ improves predictive accuracy (lower NMAE) but also increases computation time. Since the accuracy gain beyond $Q=4$ is small while the computational cost continues to rise, we thus recommend using $Q=4$ as the default setting.

---

### Official Review · Reviewer_5TkQ · 2025-11-02

**Soundness:** 3
**Presentation:** 4
**Contribution:** 3
**Rating:** 6
**Confidence:** 3

**Summary:**

This paper proposes the Multi-Output Low-Rank Nonstationary  kernel for multi-output Gaussian processes. By introducing a generalized spectral–kernel duality and modeling the matrix-valued spectral density via a low-rank Gaussian mixture, MO-LRN achieves linear scalability and superior expressiveness, outperforming existing MOGP kernels on regression and imputation benchmarks.

**Strengths:**

The paper’s strengths lie in its clear theoretical and practical contributions.

1 It establishes a new spectral–kernel duality that removes conventional restrictions and enables fully flexible matrix-valued spectral densities for multi-output Gaussian processes.

2 This paper introduces the MO-LRN kernel, a parameter-efficient yet expressive nonstationary kernel that reduces parameter growth from quadratic to linear through a low-rank spectral density with independent Gaussian-mixture factors. I really like this idea.

3 Extensive experiments validate  its effectiveness and scalability in modeling complex multi-output, nonstationary processes.

**Weaknesses:**

I really enjoy reading this paper but I am not an expert in GP. From a general perspective, I cannot find obvious flaw in this paper. I will read other reviewers' comments along with authors' feedback and ajust my score.

**Questions:**

I do not have quetions.

---

> ### Author Response · Authors · 2025-11-20
>
> > C1: I really enjoy reading this paper but I am not an expert in GP. From a general perspective, I cannot find obvious flaw in this paper. I will read other reviewers' comments along with authors' feedback and ajust my score.
>
> **Response:** We sincerely thank the reviewer for the positive assessment and for taking the time to read our paper. If you have any further questions or would like to discuss any aspect of the paper, we would be very happy to provide more details.

---

### Author Response · Authors · 2025-11-20

Dear Reviewers,

We are submitting the revised version of our manuscript and have addressed all reviewer comments. For ease of review, all modifications within the manuscript have been highlighted in blue.

Sincerely,

Authors of Paper 3121

---

### Meta-Review · Area_Chair_LMtv · 2026-01-05

**Summary:**

Three reviewers provided positive evaluations of the paper, with two of them assigning very high scores. Only one reviewer gave a score marginally below the acceptance threshold; however, the authors addressed this reviewer’s concerns by adding substantial experimental results during the rebuttal phase. Overall, the feedback on the paper is highly positive, and I therefore recommend acceptance.

**Reviewer Concerns:**

All concerns are addressed by the rebuttal.

**Reviewer Scores:**

I think Reviewer kt5w would changed his/her score if participated fully in discussion.

---

### Decision · Program_Chairs · 2026-01-26

Accept (Poster)